# ON THE RELIABILITY OF WATERMARKS FOR LARGE LANGUAGE MODELS

**John Kirchenbauer**[*1], **Jonas Geiping**[*2,3]
**Yuxin Wen**[1] , **Manli Shu**[1]**, Khalid Saifullah**[1]**, Kezhi Kong**[1],
**Kasun Fernando**[4]**, Aniruddha Saha**[1]**, Micah Goldblum**[5]**, Tom Goldstein**[1]
[1] University of Maryland
[2] ELLIS Institute Tübingen, [3] Max-Planck Institute for Intelligent Systems, Tübingen AI Center
[4] Scuola Normale Superiore di Pisa, [5] New York University

## ABSTRACT

As LLMs become commonplace, machine-generated text has the potential to flood the internet with spam, social media bots, and valueless content. *Watermarking* is a simple and effective strategy for mitigating such harms by enabling the detection and documentation of LLM-generated text. Yet a crucial question remains: How reliable is watermarking in realistic settings in the wild? There, watermarked text may be modified to suit a user's needs, or entirely rewritten to avoid detection. We study the robustness of watermarked text after it is re-written by humans, paraphrased by a non-watermarked LLM, or mixed into a longer handwritten document. We find that watermarks remain detectable even after human and machine paraphrasing. While these attacks dilute the strength of the watermark, paraphrases are statistically likely to leak n-grams or even longer fragments of the original text, resulting in high-confidence detections when enough tokens are observed. For example, after strong human paraphrasing the watermark is detectable after observing 800 tokens on average, when setting a $1e-5$ false positive rate. We also consider a range of new detection schemes that are sensitive to short spans of watermarked text embedded inside a large document, and we compare the robustness of watermarking to other kinds of detectors.

## 1 INTRODUCTION

The capability to tell the difference between machine-generated and human-written text underlies many approaches to reduce potential harms caused by generative language models (Bender et al., 2021; Crothers et al., 2022). This includes known harms, such as models being used at-scale for malicious purposes including social media bots, fake product reviews (Palmer, 2023), automated text generation on wikipedia (Woodcock, 2023), or automatic generation of targeted spearphishing attacks on vulnerable subpopulations (Schneier, 2021). Equally important, the ability to track and *document* the use of machine-generated text has the potential to reduce harms from future problems that have not yet been observed. These problems might range from the pollution of future training data (Radford et al., 2022) to the hyper-prevalence of LLM-generated blogs and other web content. Unfortunately, detection of machine-generated text is potentially difficult. Models are prompted with diverse instructions, resulting in a wide range of downstream behaviors for both machines and humans that are difficult to characterize. This can lead to low accuracy or impractical false positive rates that especially impact vulnerable subgroups, such as non-native speakers (Liang et al., 2023).

One way to enable accurate detection of machine-generated text is through *watermarking*, where generated text is marked imperceptibly so that its origin can be determined (Atallah et al., 2001; Fang et al., 2017; Kirchenbauer et al., 2023). Because watermarks rely on subtle patterns in text that are statistically unlikely to be replicated by a human, watermarking enables detectors that achieve high levels of accuracy on relatively short fragments of text. This makes watermarking a promising approach for the reliable separation of human-written and machine-generated text (Grinbaum & Adomaitis, 2022). While the effectiveness of watermarks has been shown in ideal scenarios where verbatim LLM outputs are fed directly to a detector, this is an idealized setting. In practice, humans

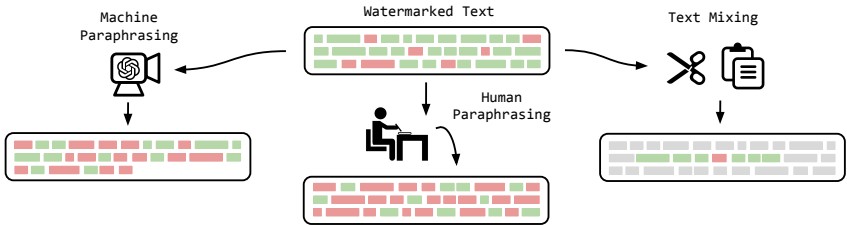

**Figure 1:** What happens to watermarked text in-the-wild? In this work we study watermark robustness against a number of text modifications, as visualized here. We visually depict that machine paraphrasing methods have a tendency to shorten texts, humans are quite effective at reducing the strength of a watermark by increasing the number of red tokens, and that short spans of watermarked text may be copied and pasted into a large document. In all of these scenarios, we find that high confidence detection reliably occurs given enough tokens as input.

may mix machine-generated text into larger documents with multiple sources. Furthermore, a human may revise or rephrase parts of the synthetic text (possibly aided by another language model) to better suit their needs, potentially even with the deliberate goal of evading detection.

In this work, we investigate the reliability of watermarking as a strategy to identify machine-generated text in realistic scenarios, based on the approach of Kirchenbauer et al. (2023). We are focused on whether watermarks remain detectable under various types of realistic corruptions (i.e. *attacks*): How reliable is watermarking when generated text is handled by humans, be it mixing with human-written text, rewriting parts or the entire passage, or feeding the text into other popular language models for rephrasing? A reliable detection strategy should be robust to these common scenarios, maintaining some statistical power and a low false positive rate (Crothers et al., 2022).

We make the following contributions:

- We re-investigate all parts of the watermark generation and watermark detection pipeline to optimize for reliability in realistic scenarios.
- We study the reliability of watermarking against paraphrasing by strong language models. When GPT-3.5 and purpose-built models are used to paraphrase watermarked text, ROC-AUC remains $> 0.85$ when $T = 200$ tokens are available, and $> 0.9$ with $T = 600$ tokens.
- We consider a "Copy-Paste" scenario where watermarked text appears inside a larger hand-written passage. When a human-written passage of length 600 tokens has 150 tokens of watermarked text inserted into it, AUC for detection is above $0.95$.
- We conduct a human study in which watermarked text is re-written by volunteers with the explicit goal of removing the watermark. While humans are relatively strong attackers, after enough observed tokens (about 800) watermarks are still usually detectable in human paraphrases even when enforcing a $1e-5$ false positive rate.
- We provide reliability estimates of watermarking compared to other state-of-the-art approaches, such as loss-based detection (Mitchell et al., 2023) and retrieval (Krishna et al., 2023), showing that these struggle at longer sequence lengths when attacked.

We argue that the correct way to characterize the strength and robustness of different detection approaches is not simply via detection accuracy metrics for a specific distribution of text, but rather to measure *how much machine-generated text* is required for each approach to succeed, and how a method behaves as a function of text sequence length. Across all the scenarios we consider in this work, we ultimately find watermarking to be more robust than other post-hoc detection methods (such as loss-based detection and caching/retrieval schemes), especially due to its favorable sample complexity, i.e., scaling behavior in terms of amount of text that is sufficient to guarantee detection.

## 2 AN OVERVIEW OF MACHINE-GENERATED TEXT DETECTION

The problem of separating human-written and machine-written text can be approached from several directions. Broadly, we can distinguish *post-hoc* detection systems that require no interaction during text generation, and *proactive* detection systems that require some action during generation. These latter systems are generally much more robust, with the downside that they have to be adopted by the model owner.

One broad class of detectors employ statistical outlier detection methods, based on entropy (Lavergne et al., 2008), perplexity (Beresneva, 2016; Tian, 2023), n-gram frequencies (Grechnikov et al., 2009; Badaskar et al., 2008), or as in DetectGPT Mitchell et al. (2023), the observation that LLMs typically assign their own text generations higher probability than "nearby" text sequences produced by span replacement with a different LLM. Even though DetectGPT exhibits superior performance compared to other zero-shot statistical outlier detection methods, it suffers from excessive computational costs. Further, all advanced statistical detectors relying on language model statistics such as perplexity or curvature ideally require white-box access to model parameters.

A retrieval-based approach for text detection, as described in Krishna et al. (2023), is a noticeably different paradigm, and an example of a *proactive* detection technique. Here, all text generated by a given model is stored in a database. Later, text samples are evaluated by matching against this database. This approach requires action taken by the model owner, but it can be quite reliable, even when faced with broad modifications of text, such as strong paraphrases.

Watermarking also requires action by the model owner as they must embed the hidden watermark signal into all outgoing text. Watermarking as a concept has a long history (Brassil et al., 1995). Older systems were based on rule-based methods to imprint watermarks into existing text (Atallah et al., 2001; Chiang et al., 2004; Venugopal et al., 2011; Topkara et al., 2006). More recent approaches for watermarking neural networks Ziegler et al. (2019); Dai & Cai (2019); Abdelnabi & Fritz (2021); He et al. (2022a;b) are learned end-to-end with both encoding and decoding of each sample. The lack of theoretical guarantees and interpretability of these approaches are problems for their widespread adoption. However, it is also possible to place watermarks on a robust mathematical foundation. Watermarks can be embedded by minimally modifying the distribution of generated output text (Fang et al., 2017; Kaptchuk et al., 2021; Kirchenbauer et al., 2023), and watermarks of this type have recently been adapted for various applications (Lee et al., 2023; Yoo et al., 2023). In this work, we mainly consider the combinatorial watermark described in Kirchenbauer et al. (2023).

## 3 HOW TO IMPROVE WATERMARK RELIABILITY?

There are a number of parameter and design choices that go into a watermark, with different parameters offering benefits in different use cases. In this section, we briefly describe the watermark proposed in Kirchenbauer et al. (2023), and the variations on the watermark that we will study.

Assume an autoregressive language model is trained on a vocabulary $V$ of size $|V|$. Given a sequence of tokens as input at step $t$, a language model predicts the next token in the sequence by outputting a vector of logit scores $l_t \in \mathbb{R}^{|V|}$ with one entry for each item in the vocabulary. A random number generator is seeded with a context window of $h$ preceding tokens, based on a pseudo-random function (PRF) $f : \mathbb{N}^h \to \mathbb{N}$. With this random seed, a subset of tokens of size $\gamma|V|$ are "colored" green and denoted $G_t$. Now, the logit scores $l_t$ are modified so that

$$l_{tk} = \begin{cases} l_{tk} + \delta & \text{if} \quad k \in G_t \\ l_{tk} & \text{otherwise.} \end{cases} \tag{1}$$

After modifications, these logit scores can be used for any desired sampling scheme. In the simplest case, one passes the scores through a softmax layer and samples from the output distribution, resulting in a bias towards tokens from $G_t$.

The watermark can be described by four parameters. The "hash" used to generate the greenlists $f$ with context width $h$, greenlist fraction $\gamma$, and the logit bias $\delta$. After watermarked text is generated, one can check for the watermark without having access to the LLM by re-computing the greenlist at each position and finding the set $s$ of greenlist token positions. The statistical significance of a sequence of tokens of length $T$ can be established by deriving the $z$-score

$$z = (|s| - \gamma T) / \sqrt{\gamma(1 - \gamma)T}. \tag{2}$$

When this $z$-score is large, one can be confident that the text is watermarked. We now discuss several variations to this scheme, which lead to improved empirical behavior.

### 3.1 Improved Hashing Schemes

The experiments in Kirchenbauer et al. (2023) focus on a simple scheme where the random number generator is seeded using $h = 1$, i.e., only a single token at position $t - 1$ is used to color the token at position $t$. We refer to this scheme as **LeftHash**. Because the greenlist depends only on one single token, a third-party observer could learn the greenlist associated with the token at position $t - 1$ by searching subsequent words at position $t$ that are less likely to appear than expected under a non-watermarked distribution. In situations where the watermark scheme is intended to be kept secret behind an API, a more secure scheme is needed.

Kirchenbauer et al. (2023) also mention a scheme (Algorithm 3) in which the greenlist at position $t$ is determined by including the token at position $t$ itself (yet to be generated), in addition to tokens to the left of $t$ in the inputs to $f$. This approach effectively increases the context width $h$ by 1, making it harder to discover the watermark rules by brute-force methods. We generalize this scheme to include arbitrary functions $f$ and text generation routines, which we describe in detail in Algorithm 1 in the Appendix. We use this method, combined with a context width of $h = 4$ and a "minimum" based method of choosing the seed token, and refer to it as the **SelfHash** scheme in the rest of the main work. For a more in-depth description of various hashing schemes and an evaluation of their performance tradeoffs, see Appendix A.5.

### 3.2 Improved Watermark Detection

The original $z$-test, Equation (2), may not be optimal when watermarked text is interspersed with non-watermarked text, eg. the case of a single paragraph of watermarked text embedded inside a much larger non-watermarked document. Because the $z$-score is computed globally over the whole document, the surrounding unwatermarked text reduces the average greenlist rate. To be able to accurately detect watermarked sub-regions even in long documents, we design a windowed test, called **WinMax** that searches for the continuous span of tokens that generates that highest $z$-score (see Appendix A.9 for a formal definition).

## 4 Evaluating Watermarking in the Wild

Watermarks are extremely accurate in simple scenarios in which a long span (50+ tokens) of text is tested in isolation and without modification. However, whether for benign or deceptive purposes, in many cases, the generated text will be embedded inside a larger document, edited by a human, or paraphrased by another language model. This section studies the robustness of the watermark under these more complex use cases.

We assume the following threat model: A user of watermarked text is aware that the text is watermarked, but has no knowledge of the hashing scheme, fraction $\gamma$, or context width $h$ that describe the watermark. They paraphrase some (possibly all) spans of the text to evade detection.

In this scenario, we can understand watermark reliability through the following two observations:

1. Without white-box access to the hashing scheme, a user cannot remove the watermark without ensuring that the re-phrased text contains none of the n-grams from the original text. If the user ever recycles long words (which often contain multiple tokens) or phrases from the original text, the watermark will remain, although with reduced strength.

2. If a paraphrased text skews even slightly toward watermarked behavior, the watermark will be detected given enough tokens. Suppose each token is only $\varepsilon$ more likely to be green than a random baseline, i.e. $|s| = \gamma T (1 + \varepsilon)$. Then for any $z$-score threshold, we expect to detect the watermark after seeing $T = (z^2 - \gamma z^2)/(\varepsilon^2 \gamma)$ tokens.

For reasons above, we do not expect paraphrasing attacks to completely remove a watermark. Rather, we expect such attacks to increase the number of tokens needed for confident detection. This hypothesis is supported by the theoretical analysis of Chakraborty et al. (2023), who assert that for an optimal detector, detection is always possible, given a sufficient number of samples.

**Experimental Setup:** Following Kirchenbauer et al. (2023), we use the Colossal Common Crawl Cleaned corpus (C4) dataset as a source of prompts for open-ended generation. For the human study, we adopt the "Long-Form Question Answering" (LFQA) dataset curated by Krishna et al. (2023)

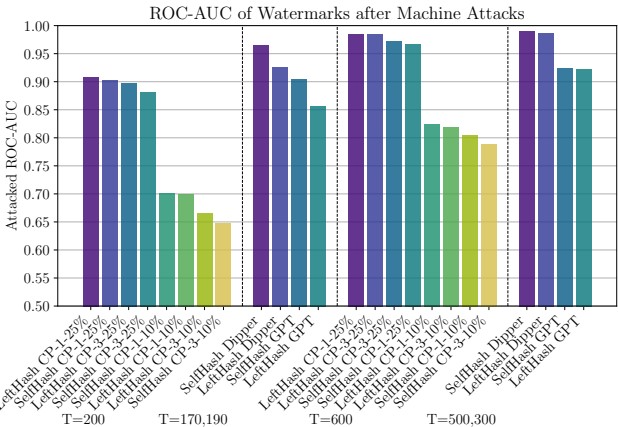

**Figure 2:** Watermark robustness to various automated text manipulations. Each bar denotes the ROC-AUC for detecting the watermark in a text that has length $T$ after the attack. Each bar label denotes the hashing scheme [SelfHash or LeftHash], and attack type [Copy-Paste (CP), paraphrase model (Dipper) and general-purpose model (GPT)]. For the Copy-Paste attack, we vary the number of watermarked passages inserted and their length. CP-3-10% denotes a copy-paste attack in which only 10% of the text is watermarked, and this watermarked text is broken across 3 different locations in the document. In the Dipper and GPT attacks, the watermarked document is re-written in its entirety, resulting in paraphrases that are shorter than the original text. In this case, the average length $T$ is reported for both GPT and Dipper, respectively.

based on a selection of posts and responses to question's on Reddit's "Explain Like I'm Five" (ELI5) forum.

In the majority of our experiments, we use `llama` (Touvron et al., 2023a), a modern state-of-the-art open source general-purpose model to generate watermarked text. In particular, we use the 7 billion parameter variant under the research use permissions of the license. In the human study portion of the experiments, we adopt `Vicuna` (Vicuna-Team, 2023), a fine-tuned variant of the base model, better suited to responding to the QA prompts used in the study. We ablate these standard dataset and model settings in the Appendix by running a set of similar experiments on alternate datasets and models.

We use a single set of language model sampling parameters across all experiments, multinomial sampling at temperature $0.7$, and for all experiments, unless explicitly stated, we use the LeftHash watermark scheme based on an additive PRF with context window $h = 1$ and $(\gamma, \delta) = (0.25, 2.0)$. This parameter combination was observed to be near the pareto frontier shown in Figure 2 of Kirchenbauer et al. (2023), i.e. extremely detectable, but with marginal cost to generation quality.

Due to the importance of text length in the performance of detection methods, including watermarking, we carefully control and specify the generation lengths considered in each experiment first by limiting the number of tokens the model can generate, and then sub-sampling the resulting data to just those generations which are within a specified range around a target length value. We use "$T$" to refer to the number of tokens considered throughout all experimental sections, and unless otherwise noted we include passages with length within $\pm 25$ tokens around that value. Unless otherwise stated, in all figures, ROC space plots and measurements are backed by $> 500$ positive and $> 500$ negative samples, and other types of point estimates are also based on $> 500$ samples.

### 4.1 ROBUSTNESS TO MACHINE PARAPHRASING ATTACKS

We run a series of *paraphrasing attacks* where we use a strong publicly available general-purpose language model API to paraphrase the text. This is a departure from the threat model of Kirchenbauer et al. (2023), who only characterize less capable models (T5). Our "GPT" paraphrase attack uses `gpt-3.5-turbo`, which is a version of the model powering ChatGPT, to rewrite the text. We also try a specially tailored paraphrasing model - the 11B Dipper model introduced in Krishna et al. (2023). We engineer the GPT prompt for optimal paraphrasing performance. Our explorations included prompts that explicitly instruct the LLM not to recycle bi-grams from the original text and we use the best performing prompt (see the Appendix for ablation details). We note that

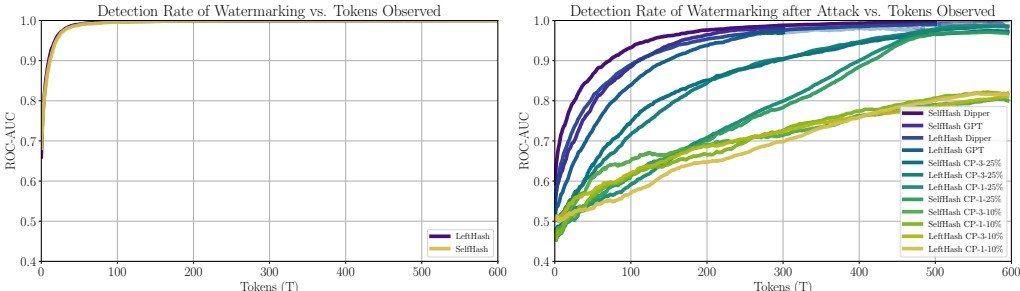

**Figure 3:** AUC as a function of text length. **(Left)** AUC of original watermarked text with no attack. **(Right)** AUC under attack. Attacks dilute the watermark strength, but the watermark recovers its accuracy as increasingly more tokens are observed. Due to the tendency of the GPT and Dipper paraphrasers to produce a shorter outputs than they receive as inputs, we make the curves translucent starting at the mean sequence length of the original text to indicate that these measurements are based on increasing fewer samples. For the Dipper attack this is $\sim 500$ and for the GPT attack this is $\sim 300$. In contrast, the Copy-Paste attack does not suffer from text shortening, and those sequences are still full length i.e. $600 \pm 25$.

when prompted for paraphrasing with longer inputs, the GPT model often effectively summarizes the text, sometimes reducing its length by more than $50\%$, which makes detecting the watermark more challenging.

The results for the main experiments attacking the watermark in this way are summarized in Figure 2. Note that we do not show the ROC-AUC numbers for the "unattacked" setting, because the detection performance of both the LeftHash watermark and the SelfHash variant at these token lengths are always $> 0.999$ for these experiments. Examining the settings where GPT or Dipper are the attack type (the smaller groups of 4 bars) with token length before and after attack of roughly $T = 200$, we see that the attack achieves a detection performance reduction of $0.05 - 0.15$ points AUC. When the lengths before attack are 600, despite the fact that Dipper and GPT reduce the lengths to 500 and 300 respectively, the attack reduces AUC by $< 0.1$ points.

Since the watermark's success and the number of tokens observed are tightly linked, we further investigate this dimension through the use of "**detectability @ T**" plots where we test prefixes of the generated sequences of lengths $T_i \in [1, ...T]$ and compute ROC-AUC for each prefix length (AUC @ T). We also provide a version of these charts where we instead show TP rates at low FPR in the appendix. Under this lens of analysis, we observe in Figure 3 that in the unattacked setting, the AUC @ T quickly approaches its eventual value of $\sim 1.0$. In the attacked setting, we see that the AUCs are reduced overall by all methods, but that despite how successful a paraphrasing attack might look at 200-300 tokens, by the 600 token mark, under the GPT and Dipper model-based attacks, the watermark recovers to an AUC greater than 0.9.

## 4.2 ROBUSTNESS TO COPY-PASTE ATTACKS

We devise a synthetic but realistic attack where we take watermarked text fragments and embed them inside a surrounding un-watermarked, human-written document. This creates heterogeneous examples where only a few sub-spans of the text are watermarked. The attack method has two parameters, (1) the number of watermarked span insertions and (2) the fraction of the resulting document that represents watermarked text. For example, consider a passage with $10\%$ watermarked tokens and 3 insertions. If the original text is 1000 tokens, then this means that 3 watermarked spans of 33 tokens would be inserted into the enclosing chunk of human text. We give this setting the short name "CP-3-10%," which is an abbreviation of "Copy-Paste with 3 spans at 10% watermarking."

Returning to Figure 2, we see that with $25\%$ watermark remaining, the copy-paste attack procedure has a much stronger effect on the watermark than the other two machine based attacks, dropping the AUC to below 0.7 for 200 tokens and below 0.85 for 600 tokens. Examining Figure 3 we congruently see that the watermark detectability grows more slowly than in the unattacked setting, but grows steadily nevertheless. We revisit this reliable growth behavior in Section 4.4 where we compare watermarking to alternative detection methods.

### 4.3 PARAPHRASING BY HUMAN WRITERS

Humans may paraphrase watermarked text to better fit the style or tone of an existing passage or to better suit the tastes of the user. These modifications are a key part of every-day interactions that human users have with generated text, and to be reliable in-the-wild, a watermark should be robust to these changes. However, in a more adversarial setting, a human may also paraphrase text with the deliberate goal of evading detection.

To test the feasibility of evasion, we set up a human study. We recruit 14 experienced human writers (graduate students) who were presented with watermarked text. They were asked to paraphrase the text with the goal of removing the watermark while attempting to maintain passage length and semantic content. In addition to compensation for participation, top performers were awarded one of three $100 gift certificates to incentivize strong but faithful paraphrasing. Each human writer is given a different set of text passages, which we generate by prompting a watermarked version of Vicuna with the LFQA data.

We first validate that all human writers successfully paraphrased the text passages using P-SP (Wieting et al., 2022) scores. Doing so, we find P-SP scores far exceeding the threshold of 0.7 considered an expert paraphrase in Wieting et al. (2022). We show this score for each writer in Figure 4 (left), comparing P-SP directly to the $z$-score of detection based on their text. We then analyze the detectabilty of the watermark via $z$-scores in the right plot of Figure 4. Here, we show watermark strength as a function of $T$, i.e. of tokens seen during detection. While human writers are strong paraphrasers, they cannot escape the two observations posed in Section 4. As shown in Figure 4 (right), eventually, the evidence for a watermark mounts and even human writers are clearly detected on average after 800 tokens.

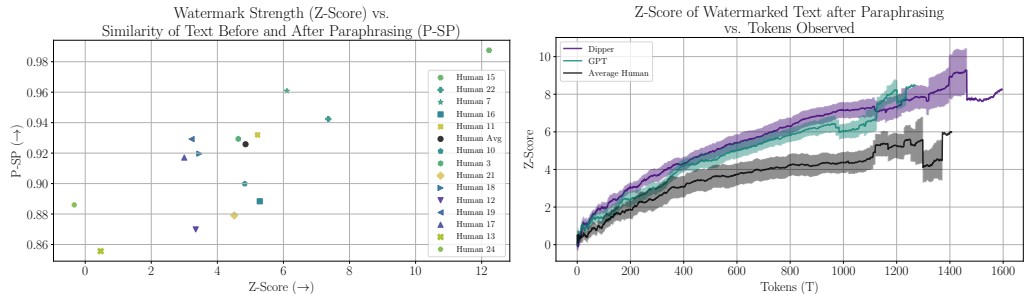

Figure 4: **(Left)** Paraphrase quality for human writers as measured by P-SP. Note the y-axis. **(Right)** Summary results for each paraphrasing attack in aggregate, showing that human writers are stronger paraphrasers than both machine attacks. Yet, all attacks can be detected with certainty after 400 to 800 tokens.

### 4.4 A COMPARATIVE STUDY OF DETECTION ALGORITHMS

A number of alternatives to watermarks exist. As discussed in Section 2, other paradigms for LLM detection are post-hoc detectors based on statistical features, black-box learned detectors, and retrieval systems. Based on previous work in Mitchell et al. (2023), we study DetectGPT as a representative for loss-based post-hoc detectors, RADAR (Hu et al., 2023) for detectors based on a learned classifier, and we use the retrieval method proposed by Krishna et al. (2023). We briefly describe the methods below and provide further details and hyperparameters in Appendix A.11.3.

**Retrieval:** A retrieval approach to detection requires the creation and maintenance of a comprehensive database of all sequences previously generated by the language model to check against at test time and we adopt the retrieval method of (Krishna et al., 2023), utilizing the BM25 search method since it performed the best in their experiments.

**RADAR:** To develop a learned classifier approach that is robust to paraphrasing, Hu et al. (2023) propose RADAR, a framework for training a robust AI-text detector using adversarial learning. We use the publicly available model, RADAR-Vicuna-7B, where the discriminator has been robustly optimized to detect Vicuna generations. Due to 512 token limit of the model, all sequences are always truncated to that length even if they are longer.

**DetectGPT:** DetectGPT (Mitchell et al., 2023) is a post-hoc method for detecting machine-generated texts that employs a curvature-based criterion which compares the log probabilities of a candidate passage and perturbed versions of the same passage. We use the official implementation of DetectGPT with its default setting of `T5-3B` (Raffel et al., 2020) as the mask-filling model and utilize their best performing hyperparameter settings.

**Which method is most reliable?** We take these alternate detection approaches and attack them with the battery of machine attacks described in the previous section. We plot ROC charts for each detection scheme when evaluated on attacked texts in Figure 5. We find that retrieval and watermarking are both decent detection methods, RADAR is reasonably effective in the machine paraphrase settings, and all are significantly more reliable than DetectGPT. Yet, while retrieval equals or outperforms watermarking on the GPT and Dipper model based paraphrases (corroborating the results in Krishna et al. (2023)), it and RADAR are bested by watermarking in the copy-paste setting when analyzing performance in the range of FPR < 0.1.

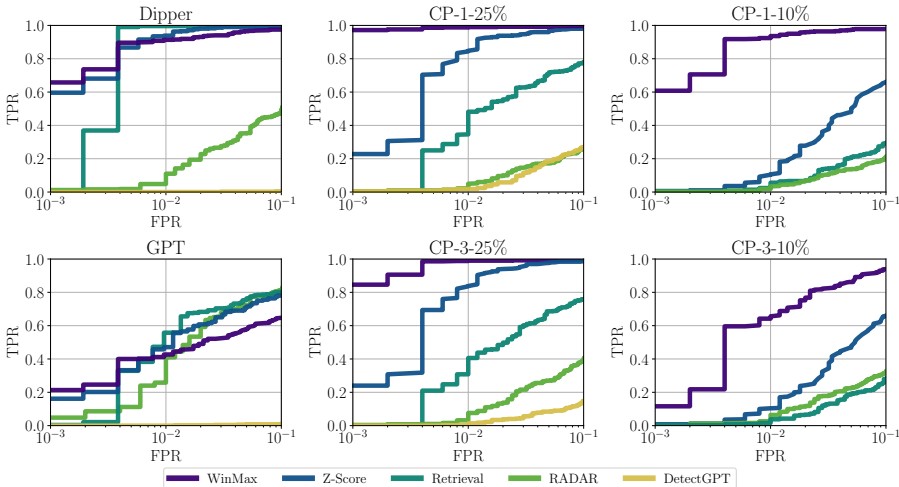

**Figure 5:** ROC charts for various machine attack types for a text passage with length before attack of $T = 1000$. For Dipper and GPT, the length after attack is decreased to 800 and 300 respectively. We focus on the low FPR region by log scaling and limiting the x-axis to the interval $[0.001, 0.1]$. $z$-score and WinMax denote the two watermark detectors, "Retrieval" is the method of Krishna et al. (2023). **(Left Column)** The two machine paraphrase attacks, **(Center Column)** The copy-paste attack with a remaining detectable text percentage of $25\%$ spread over 1 and 3 segments. **(Right Columns)** The copy-paste attack at only $10\%$ remaining detectable text. While DetectGPT performs poorly in all cases, Retrieval is quite robust to the machine paraphrase attacks, and RADAR performs well against the GPT paraphrase attack. However, watermarking outperforms both Retrieval and RADAR under the copy-paste attack, with the WinMax detector variant performing significantly better at the most severe copy-paste settings.

**The Sample Complexity of Various Detection Schemes.** To hone in on the observed differences between the schemes under each attack setting, in Figure 6 we also compare all three approaches in terms of AUC @ T, i.e. detection performance as a function of text quantity. We note the details of how Retrieval, DetectGPT, and RADAR were evaluated at varied values of $T$ in the appendix.

Here, we start to develop an explanation for why watermarking outperforms other methods in this specific setting by observing that *scaling behaviors for each detection method differ starkly under attack*. While all approaches grow in sensitivity as a function of T in the weaker setting, under strong attacks, only the WinMax watermarking scheme reaches desired levels of reliability. In particular, when only a small fraction of text (CP-X-10%) remains under the copy-paste attack, non-watermarking methods struggle as text length increases. For the Retrieval and RADAR methods, as $T$ increases, the ratio between tokens coming from the original unattacked text (positive), and tokens in the surrounding text (negative), decreases. Therefore, the overall similarity to the original example continues to drop, which eventually causes a decrease in performance for both copy-paste attack severities. For DetectGPT, the same effect is observed but at more drastic levels. We investigate these non-monotonic behaviors in more detail in Appendix A.12 by examining trends in the actual Retrieval and DetectGPT scores produced under each attack setting.

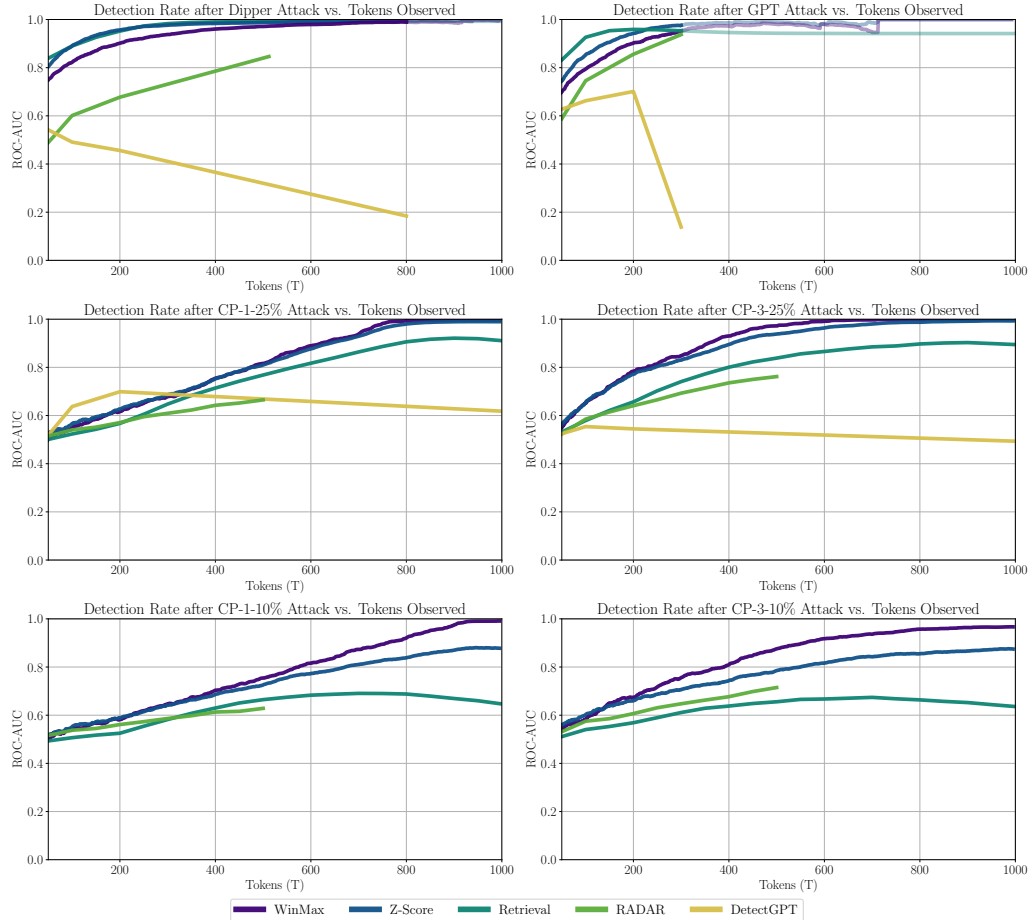

**Figure 6:** AUC at $T$ for various types of attack. **(Top Row)** The Dipper and GPT machine paraphrasing attacks. **(Center Row)** The copy-paste attack with a remaining detectable text percentage of 25% spread over 1 and 3 segments. **(Bottom Row)** The copy-paste attack at only 10% remaining detectable text. While all schemes are impacted by the various attacks, DetectGPT scales particularly poorly under attack. Retrieval and RADAR both show a positive trend in detectability as a function of T, but their growth is slower or non-monotonic, whereas watermarking steadily improves in power for all attacks evaluated. Due to the tendency of the GPT and Dipper paraphrase models to produce a shorter sequence of tokens as output than they receive as input, we make the curves translucent starting at the mean sequence length of the attacked texts to indicate that these measurements are based on increasingly fewer samples and are therefore more uncertain. For DetectGPT and RADAR, the last measurement for each attack type is just computed on the set of full sequences (rather than rolling as with watermarking), and so we plot the result at that average T value. For the Dipper attack this is $\sim 800$ and for the GPT attack this is $\sim 300$. Due to the synthetic nature of the Copy-Paste attack, after attack, those sequences are still full length i.e. $1000 \pm 25$. However, for RADAR, we always only analyze up to $T = 512$ due to the model's context limit.

## 5    CONCLUSIONS

Through a comprehensive empirical investigation, including strong machine paraphrasing and human writers, we evaluate the reliability of watermarks as a mechanism for the documentation and detection of machine-generated text. We advocate for a view of watermarking reliability as a function of text length, and find that even human writers cannot reliably remove watermarks if being measured at 1000 words, despite having the goal of removing the watermark. This view of watermark reliability as a function of text length turns out to be a strong property of watermarking. When comparing to other paradigms, such as retrieval and loss-based detection, we do not find a strong improvement with text length, making watermarking out to be the most reliable approach in our study.

## 6 REPRODUCIBILITY STATEMENT

In service of reproducibility, we have included all key details necessary to understand our experimental setup. In sections of the main work where space was at a premium, we included pointers to sections in the Appendix that provide further information. All models and data sources utilized are publicly available and the compute infrastructure used was based on commodity-level CPUs and GPUs running open source software. The implementation is designed to be both useable and extensible for further research and can be found at github.com/jwkirchenbauer/lm-watermarking.

## 7 ETHICS STATEMENT AND DISCUSSION OF SOCIETAL IMPACTS

In this work we conduct a human study where we collect written annotations and preference determinations from study participants. Before conducting this portion of the research, we submitted a proposal to our Institutional Review Board office and received an "Exempt" status. We include all details of the human study setup and evaluation procedure as well as a description of the compensation provided to participants. Additionally, the general topic of AI-generated content detection is an important domain of study with regards to societal trust and safety especially in online spaces. We conduct this research with an awareness of the impact that results on these topics can have if misrepresented and are careful to report our findings in an accurate and impartial manner, and we include a careful discussion of societal implications below.

While our work focuses on technical aspects of the reliability of machine-generated text detection, these findings have further societal implications. A strong finding that our study supports is that the problem of detection is simply made *tractable* through watermarking. Compared to other approaches, we find watermarking to be most reliable in everyday scenarios, where text is taken from a generative model, modified by human writers or other models and inserted into spans of human-written text. *This suggests that watermarking may be a promising strategy to reduce harm arising from the use of large language models.*

In pursuit of harm reduction through technology like watermarking, it is crucial to consider several precise details. First, from previous studies, i.e. Kirchenbauer et al. (2023), we know that watermarks can be broken by sufficiently motivated attackers, e.g. using generative attacks. Nevertheless, our results among others show that watermarks can be used to reliably track machine-generated text as it distributed over the internet in non-adversarial scenarios. From this perspective, watermarking technology flips the problem on its head: whereas before, machine-generated texts were mostly undetectable by default, with advanced watermarks, active effort is required to make the text undetectable. Second, while technologies like watermarking will not be *perfect*, to be deployable in the real world, detection strategies must guarantee low false-positive rates, which have to be weighed against detection sensitivity, as every false positive can be extremely harmful. Other approaches like some of the post-hoc detectors we study cannot consistently provide low false positive rates in all domains, and thus can fail, for example when classifying text from non-native speakers. Third, whether harm is reduced is scenario-dependent. For example, in classroom settings, a false accusation is serious, whereas even positive detection results have only minuscule benefits to the learning experience of students. On the other hand, solutions that detect fake news and spam on social media platforms that might include watermarks as one additional feature in a larger system, might trade relaxed error tolerances for clearer expectations through terms of service that communicate both its utility and potential limitations.

## 8 ACKNOWLEDGEMENTS

This work was made possible by the ONR MURI program, DARPA GARD (HR00112020007), the Office of Naval Research (N000142112557), and the AFOSR MURI program. Commercial support was provided by Capital One Bank, the Amazon Research Award program, and Open Philanthropy. Further support was provided by the National Science Foundation (IIS-2212182), and by the NSF TRAILS Institute (2229885).

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

# A  APPENDIX

We provide a number of extended sets of visualizations and ablation studies to supplement the results shown in the main body of the work as well as more methodological and experimental details. Below is a table of contents to help navigate the various subsections.

**Table of Contents**

## A.1 WHAT ABOUT THE WHITE-BOX SETTING?

In this work we have focused on the black-box setting, where text is modified in plausible use cases for machine-generated text and the secret key of the watermarking scheme is unknown to the party modifying the text through paraphrasing or other editing.

Once this assumption is relaxed, for example if the secret key is breached or the watermark is public, then an attacker with white-box access to a strong paraphrasing model like Dipper (Krishna et al., 2023) could break the watermark in the following way: The attacker can apply an *anti-watermark* scheme during generation with the paraphrasing model, where instead of adding $\delta$ in Equation (1) to logit outputs, $\delta$ is instead subtracted. The attacker can further keep track of the current score of green-listed versus red-listed tokens and can accordingly modify this negative $\delta$ on-the-fly to guarantee that of the $T$ tokens of text generated by the paraphrasing model exactly $\gamma T$ are colored green, and no watermark signal is leaked into the paraphrased text.

This attack relies on both white-box access to the watermark key and availability of a strong paraphrasing model. To bypass this difficult threat model, the watermark context width $h$ has to be chosen sufficiently large so that in an adversarial scenario, the watermark key could not be easily discovered (or alternatively, sufficiently many keys must be employed simultaneously, as discussed in Kirchenbauer et al. (2023)). Nevertheless, *not all watermarking use cases are necessarily adversarial*, and we strongly believe that in benign cases, even a watermark with imperfect resistance to text corruption is still very valuable. Documenting the usage of machine-generated text overall, for example either to trace the spread of generated text in hindsight, or to remove generated text from future training runs (Radford et al., 2022; Shumailov et al., 2023), can provide a baseline effort to "future-proof" popular generative models.

## A.2 Relationship to Theoretical Results on (Im)Possibility of Detection

Several works have investigated the difficulty of detecting language models from a theoretical (Varshney et al., 2020; Sadasivan et al., 2023; Chakraborty et al., 2023) and practical (Bhat & Parthasarathy, 2020; Wolff & Wolff, 2022; Tang et al., 2023) perspective, although these works mostly pertain to post-hoc detectors. How does this body of work relate to our empirical evidence on the reliability of watermarking?

In their work on the impossibility of LLM watermarks and detection, Sadasivan et al. (2023) assume the goal of detection is to discern text generated by a large language model from text generated by a randomly sampled human from some population, for example tweets from Twitter users. Sadasivan et al. (2023) also assume that the goal of large language model training is to mimic such a human language distribution. If the LLM perfectly mimics this distribution, then its samples will be indiscernible from the human generations. To the extent that the LLM imperfectly mimics the human distribution, Chakraborty et al. (2023) prove that the distribution shift between the language model and human-written text can be detected, given a sufficient number of samples. If an LLM instead followed a more concentrated distribution over text, for example by routinely speaking in the voice of a particular individual rather than a randomly sampled one, then its samples would be unlikely under the generic human distribution and would therefore be detectable.

Existing literature suggests that LLMs trained with standard methods do not mimic randomly sampled individuals – the standard classification loss used to train LLMs is known to reward a low entropy output distribution that is distinct from typical human text (Holtzman et al., 2019; Gehrmann et al., 2019). Furthermore, benign users, and by extension companies that sell LLMs as a service, seldom want an LLM to mimic a generic human language distribution, and may instead prefer inhumanly understandable and factual text, or text in a polished professional voice.

Regardless of whether future LLMs mimic a human distribution, watermarking is still possible. Consider that a generic human language distribution is incredibly diffuse, with many valid completions for a single prompt. For example, different mathematicians write unique yet correct proofs of the same theorem, even if their logical steps are identical, indicating that even tasks with seemingly narrow answers, like math, may actually still admit diffuse human language distributions.

The high entropy over completions enables watermarks to concentrate model outputs on a subset of valid completions while still spreading mass across diverse and high-quality candidates. In fact, watermarks which only make very small changes to the generative distribution can be detected with enough samples, as proved theoretically in Chakraborty et al. (2023) and verified empirically in our own work. Moreover, the benefit of watermarking is that we can minimally change the generative distribution in a way that optimizes detectability.

Sadasivan et al. (2023) also state that, in principle, a watermark can be removed by a paraphraser that samples from the set of text with the same content. Such theoretically optimal paraphrasers have so far not been demonstrated. Our experiments show that even stronger models (chatGPT) may be insufficient for paraphrasing weaker models (LLaMA-7B), demonstrating that watermark detection is possible when contending with the paraphrasers that exist today.

Ultimately, the theory literature discussing differences between language distributions does not get at the core of the detection problem. Existing post-hoc detectors do *not* fail because the distributions of human-written and machine-generated text are so similar, but instead because we lack a mathematical *characterization* of their differences (Liang et al., 2023; Krishna et al., 2023). Consider, for example, an LLM with a pseudo-random sampler and a fixed seed. Each token is a deterministic function of those that came before it. The output distribution is concentrated on a single example conditional on each prompt, making it very different from a human distribution. Yet, without white-box model access, detection is still difficult in practice because we know neither the location of this peak for the LLM nor the distribution of human text. In contrast, watermarking is effective not because the distributional shift it induces is large, but because this shift is characterized by a simple rule. The human-written and machine-generated text distributions were likely very far apart before watermarking, but a characterization of the differences is needed for detection.

### A.3    ON THE RELIABILITY OF P-VALUES

In their original work, Kirchenbauer et al. (2023) discussed the impact that repeated n-grams in a given piece of text would have on the validity of the independence assumption of the z-test. While we do not utilize p-values in our analyses, and instead consider empirical FPR and TPR rates, we run some experiments to evaluate the significance of the impact that counting or not counting repeated n-grams at detection time can have . Concurrently, the work of Fernandez et al. (2023) presents a more in depth analysis of this effect on the watermark of Kirchenbauer et al. (2023) as well as other watermarking techniques with detection algorithms based on hypothesis tests. We find that while the p-values produced by the uncorrected version of the detection test suggest a lower FPR than we actually measure at a given z-score threshold, the correction for n-gram repeats improves the correspondence between the empirical FPR and the p-value significantly.

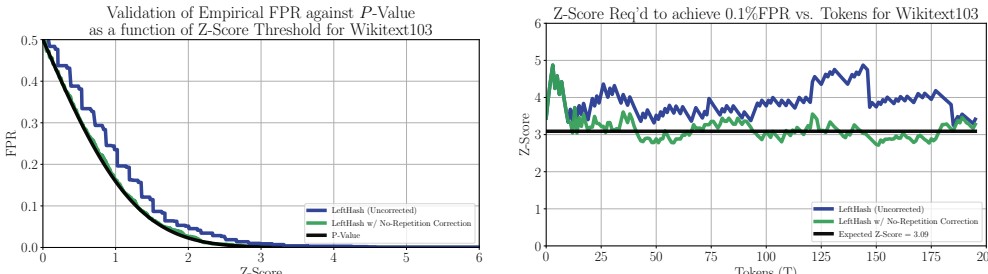

**Figure 7:** The **left** chart shows the observed FPR as function of different z-score cuttoffs along with the p-value (in black) corresponding to a one-tailed z test. The other curves show that the basic LeftHash scheme (blue) produces a slightly higher FPR than the corresponding p-value, but that the correction based on not counting repeated n-grams during detection (green) fixes the error underestimation. The **right** chart shows the Z-score threshold required to achieve a FPR of $0.1\%$ as a function of the length of the text in tokens. For small values of T, the normal assumption is slightly violated, as the z-score required is closer to $4$ whilst the z-score that should produce a p-value of $0.1\% = 0.001$ is actually 3.09 (black). As the sequence gets longer we see that the threshold required by the uncorrected version of the LeftHash (blue) scheme remains above the analytical value, but that correcting by not counting duplicate n-grams at detection time (green) causes the z-score threshold to correctly settle around the expected value of 3.09

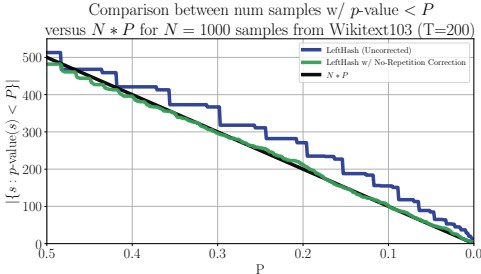

**Figure 8:** A plot of the correspondence between the empirical fraction of sequences whose p-value is lower than a given probability P, and the quantity $N * P$. While the uncorrected version of LeftHash incurs a false positive rate (y axis) slightly larger than $N * P$ given a threshold $P$, the correction of not counting repeats fixes this issue.

A.4    UTILIZING BETTER QUALITY METRICS

To more accurately examine the effects of watermarking, in addition to utilizing a stronger generative model from the `llama` family (Touvron et al., 2023a), we employ a pair of metrics designed to capture different aspects of generation quality. Given the fraction $u_n$ of unique $n$-grams in a sequence of text, we define text diversity up to order $N$ via

$$\text{diversity} = -\log\left(1 - \prod_{n=1}^{N} u_n\right), \tag{3}$$

to represent a view on n-gram repetition metrics described in Welleck et al. (2019) and Li et al. (2022a) in a more readable format. A higher diversity score represents a more diverse text, where fewer $n$-grams are repeated.

To estimate whether watermarked text drifts away from un-watermarked model generations, we adopt the same evaluation metric as Krishna et al. (2023) to measure paraphrase similarity: P-SP (Wieting et al., 2022). We measure similarity between different sets of text pairs such as un-watermarked and watermarked outputs, or the human gold completion to a prompt versus the watermarked completion generated by the model. Further, we evaluate human annotator judgement of un-watermarked versus watermarked text quality (Table 5).

We also considered other metrics for language model sampling quality such as MAUVE (Pillutla et al., 2021) and coherence as described in (Su et al., 2022; Li et al., 2022b). However, the insight those measures provided when comparing outputs under various generation and watermarking settings was generally subsumed by what the P-SP metric revealed, so we chose to simplify presentation using just P-SP as our semantic similarity metric across all relevant visualizations. Aside from the study of watermarks, we note that output quality evaluation in open ended generation settings is still a research area with many open questions.

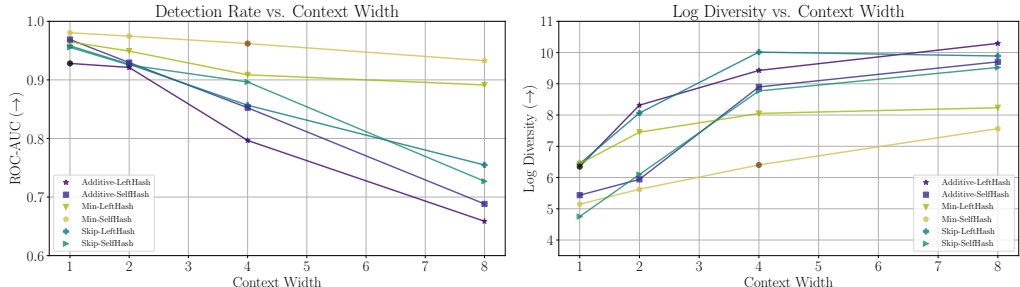

**Figure 9:** Effect of context width on watermark robustness and diversity. **(Left)** Effect of context width on watermark robustness as measured by ROC-AUC after a paraphrasing attack by GPT. For larger context widths `Skip` and `Min` variants provide the best detection strength. **(Right)** Effect of the seeding scheme context width on the quality of the text as measured by log diversity. A small context width produces less diverse outputs for all three schemes, and the `Additive` and `Skip` schemes produce more diverse text at larger context widths than the `Min` scheme. **(Both)** Watermark parameters $\gamma, \delta$ are fixed at $(0.25, 4.0)$. The black circle marks the simple `Additive-LeftHash` with context width $h = 1$ scheme, and the brown circle marks the width $h = 4$ variant of the `Min-SelfHash` scheme, both evaluated throughout the work (names shortened to "LeftHash" and "SelfHash" respectively).

## A.5    HASHING SCHEME EXTENDED ABLATION

When the context width $h$ is increased to maintain secrecy of the red/green list rules, we find that detection reliability substantially depends on the hashing scheme. We define the following functions $f : \mathbb{N}^h \to \mathbb{N}$ that map a span of tokens $\{x_i\}$ onto a pseudo-random number. Each depends on a secret salt value $s \in \mathbb{N}$ and a standard integer PRF $P : \mathbb{N} \to \mathbb{N}$.

**`Additive`**: This is the function described in Kirchenbauer et al. (2023). We extend it to $h > 1$ by defining $f_{\text{Additive-LeftHash}}(x) = P\left(s \sum_{i=1}^{h} x_i\right)$. While permutations of the context $x$ do not change the outcome, removing or swapping a single token from $x$ changes the hash and hence breaks the watermark at this token.

**`Skip`**: This function uses only the left-most token in the context: $f_{\text{Skip-LeftHash}}(x) = P(sx_h)$. This hash is robust to changes in the non-leftmost token, but it is susceptible to insertions/deletions.

**`Min`**: This function is defined by $f_{\text{Min-LeftHash}}(x) = \min_{i \in 1,\ldots,h} P(sx_i)$. It is robust to permutations within the context and it is partially robust to insertions/deletions. Given that all $P(sx_i)$ are pseudo-random and equally likely to be the smallest value, the likelihood of failure of this scheme is proportional to the number of values removed from the context, i.e. if $h = 4$ and 2 tokens are removed/missing from the context, the PRF is still $50\%$ likely to generate the same hash.

**Choosing a Scheme.** Figure 9 shows that a small context width $h$ provides the best robustness to machine paraphrasing. At wider context widths, `Skip` and `Min` variants remain strong under attack while `Additive` suffers. However, we see that this robustness improvement comes at a trade-off to text quality as the `Min` schemes produce less diverse outputs. Still, at a context width $h = 4$, the `Min-SelfHash` scheme (brown circle marker) achieves the same diversity as the original `Additive-LeftHash` scheme at width $h = 1$ (black circle), while being more robust. This shows that we can use the additional strength provided by `Min` and `SelfHash` to run longer context widths, which in turn secure the watermark. We adopt these two schemes as "SelfHash" and "LeftHash" respectively in the sections of the main work.

In this section, we complete our extensive study of watermark hyperparameters by presenting a representative selection of settings varying different components of the hashing scheme to explore different parts of the pareto space between watermark strength and text quality. We further detail our proposed extension of the `SelfHash` algorithm to arbitrary sampling schemes and pseudorandom functions $f$ in Algorithm 1.

When using greedy decoding, the scheme in Algorithm 1 covers the scheme denoted as Alg.3. in Kirchenbauer et al. (2023). We also note that in practice, iterating over all $k \in |V|$ is not strictly

---

**Algorithm 1** Generalized `SelfHash` Watermark

---

**Input:** Context $x_h, \ldots x_1$, vocabulary $V$, arbitrary text generation scheme $S$, LLM logits $l$
       Watermark hyperparameters $\gamma \in (0, 1), \delta > 0, f, h > 0$, integer hash $P$

$G = \emptyset$                 ▷ Initialize empty set of green-listed tokens
**for** $k = 1, \ldots, |V|$ **do**
    $H_k = f(x)P(k)$              ▷ Compute $k$-th key
    $G_k = \text{RandPerm}_{H_k}(V)[: \gamma |V|]$     ▷ Temp. green list $G_k$ seeded with $H_k$
    **if** $k \in G_k$ **then**
        $G \leftarrow G \cup \{k\}$        ▷ Include $k$ in final green list if self-consistent
    **end if**
**end for**

$$l_k \leftarrow \begin{cases} l_k + \delta & \text{if} \quad k \in G \\ l_k & \text{otherwise} \end{cases}$$

Sample a new token $x_0$ from modified logits $l$ using sampling scheme $S$.

---

necessary. We only iterate over the 40 indices $k$ with the largest logit score $l_k$ and skip all others, for which the addition of $\delta$ is unlikely to make an impact on the final probability distribution.

We further note that the choice to set $H_k = f(x)P(k)$ instead of of $H_k = f([x, k])$ in Algorithm 1 is not strictly necessary. We refer to the first choice as `anchoring` and ablate it separately in Figure 12, where $H_k = f([x, k])$ denotes the un-anchored approach. On all other occasions the SelfHash scheme is always anchored as described in Algorithm 1.

In Figure 10, to vary the strength of the watermark, we test a selection of $\gamma$ and $\delta$ values in $\{0.5, 0.25, 0.1\}$ and $\{1.0, 2.0, 4.0\}$ respectively. These values give us 9 settings representing both weak and strong watermarks that yield a series of increasing $z$-scores across the $x$-axis for each seeding scheme. All points in these charts are averages computed on $\sim 500$ samples with token length $T = 200 \pm 25$.

We observe that as the watermark is made stronger (resulting in higher $z$-scores) certain schemes trend positively with respect to $n$-gram diversity of their outputs and others trend negatively. Namely, for the "Additive-LeftHash,1" (i.e. using `LeftHash` with context width $h = 1$, and additive $f$) scheme that was evaluated in Kirchenbauer et al. (2023), stronger watermark strength yields less text diversity. This issue is remedied by using a larger context width, or by choosing the "Skip-LeftHash,4" scheme which exhibits improved text diversity at higher watermark strengths. This finding provides another advantage for schemes with increased context width.

In Figure 11, we display the drift between unwatermarked and watermarked completions, as measured in P-SP. To put these numbers into perspective, we note that the drift between human text and unwatermarked text can be estimated as just under $0.50$ PSP. This implies that for the watermark settings yielding $z$-scores up to 10.0 (a score that is *extremely* detectable), the semantic divergence of the watermarked output from the model's unwatermarked behavior is less than the average divergence of the unwatermarked model from human gold completions.

In Figure 9 from the main body we empirically demonstrate that the attack amplification effect, hypothesized in Kirchenbauer et al. (2023) to occur when using watermark schemes with context widths greater than $h = 1$, is made less severe when using `Skip` and `Min` based schemes. To get the full picture based on all degrees of freedom afforded in the hashing scheme space described in the main body, we provide a plot varying all parameters simultaneously in Figure 12. We confirm the prediction in Kirchenbauer et al. (2023) that the generalized version of the SelfHash algorithm, when applied to the Anchored version of MinHash scheme at a moderate context width (i.e "Algorithm 3" from the previous work when the context width is $h = 4$), provides a competitive tradeoff between robustness to attack and text diversity and quality along with the added benefit of being more secure against reverse engineering of the PRF key than the Additive-LeftHash.

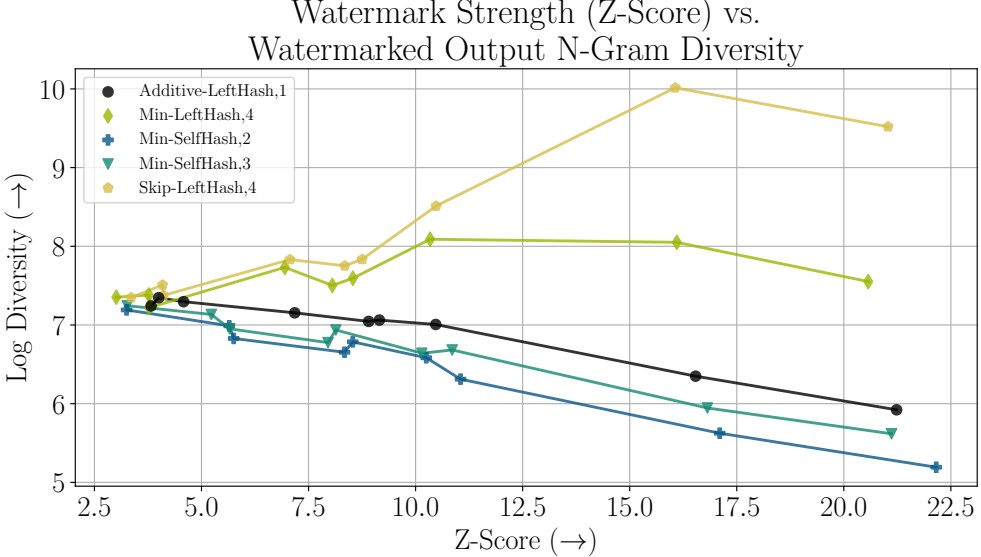

**Figure 10:** The pareto frontier with watermark strength ($z$-score) on the $x$-axis and text diversity (Log Diversity, see Appendix A.4) shown on the $y$-axis. Higher is better for both metrics. We see that the Min-LeftHash and Skip-LeftHash schemes with larger context windows yield more diverse generations as watermark strength increases. The standard LeftHash scheme with context width 1 and the SelfHash based schemes produce lower diversity outputs under strong watermarking.

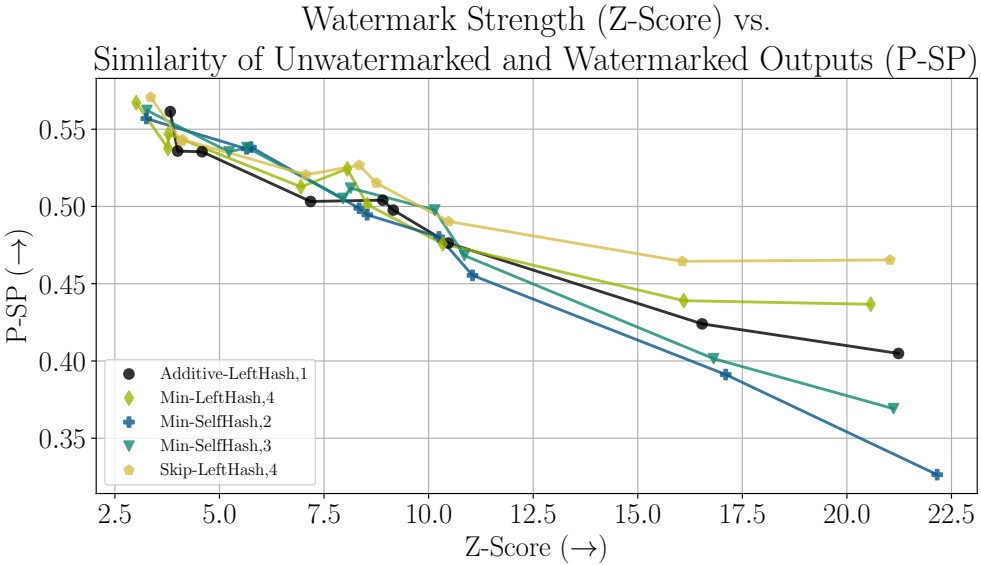

**Figure 11:** The pareto frontier of watermark strength in $z$-score, shown on the $x$-axis, versus similarity between the un-watermarked and watermarked output, shown on the $y$-axis. Higher is better for both metrics. We see that the Min-LeftHash and Skip-LeftHash schemes with larger context windows produce watermarked generations with slightly higher semantic similarity to their unwatermarked counterparts than the other schemes as watermark strength increases. However for all schemes, especially the SelfHash based settings, as the watermark is made stronger, the watermarked output diverges more from the unwatermarked output.

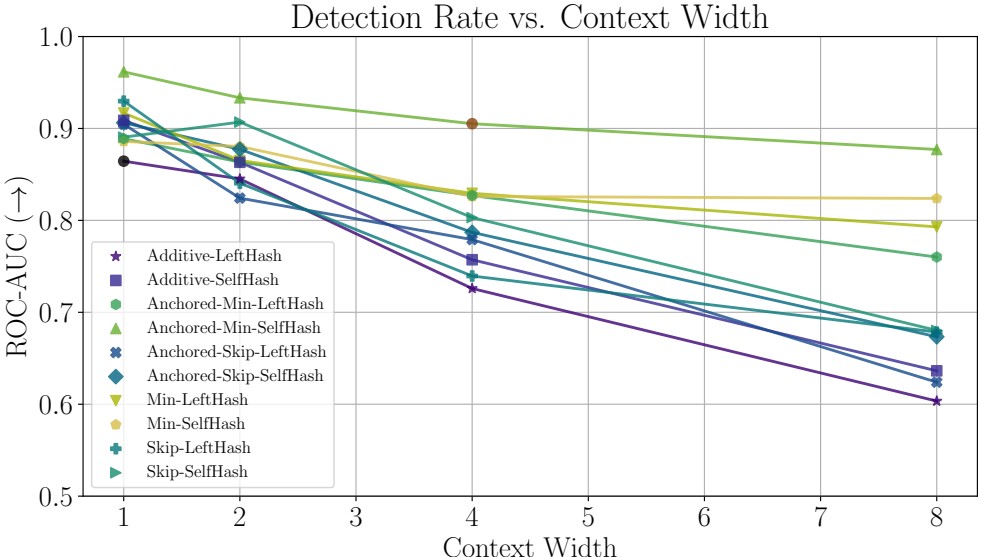

**Figure 12:** Effect of seeding scheme context width on watermark robustness as measured by ROC-AUC after a paraphrasing attack by GPT. In this variant, we ablate two more parameters, "anchoring" and "self-salting" (SelfHash). The watermark strength parameters $\gamma, \delta$ are fixed at $(0.25, 2.0)$, slightly weaker than the above to bring in line with settings from figures in main work. We specially denote "Additive-LeftHash" at width 1 via the black circle marker and also the "Anchored-Min-SelfHash" at width 4 in as a brown circle, as these correspond to the "LeftHash" and "SelfHash" variants respectively shown in Figure 2, where we also reported the more complex watermark seeding scheme outperforming the standard scheme proposed by Kirchenbauer et al. (2023). Here, it becomes clear that this is the result of the Anchored-Min-SelfHash scheme outperforming the others in terms of robustness at all context widths.

## A.6 EFFECT OF WATERMARKS ON UTILITY: TRIVIAQA

Kirchenbauer et al. (2023) performed a small study on the impact of their watermark in knowledge intensive, factuality critical generation scenarios but only evaluated the impact of the LeftHash scheme. While we do not expect to see major differences in the impact on utility for the SelfHash scheme (only improvements in reliability) for completeness, we explicitly verify this.

For this experiment, we generate free-form answers to the first 1000 TriviaQA questions. We use the slightly more advanced chat models LLaMA-2-chat-7b (Touvron et al., 2023b) and Zephyr-7b-beta (Tunstall et al., 2023), a finetuned version of Mistral-7B (Jiang et al., 2023). We score the responses as correct if the string of the correct answer or one of its aliases are *included* in the model's response. We found the responses of the chat models to be too verbose to get consistent "Exact Match" scores reflective of their capability. The results are shown in Table 1.

Overall, we find that the utility of the model is only minimally impacted by the SelfHash watermark scheme, with the original LeftHash watermark yielding similar results. This is in line with our expectations that the two schemes should behave similarly under this type of analysis. As shown in Kirchenbauer et al. (2023), the strength of the watermark is quite low in these short, low-entropy generation scenarios, with the watermarked responses only producing marginally more significant detection statistics than the unwatermarked generations.

| Watermarking Scheme | None | LeftHash | SelfHash |
|---|---|---|---|
| Llama2-7B-chat | 0.523 | 0.517 | 0.524 |
| Zephyr-7B-beta | 0.616 | 0.612 | 0.607 |

Table 1: **(Performance)** TriviaQA performance under the "generation *includes* target?" scoring method.

| Watermarking Scheme | None | LeftHash | SelfHash |
|---|---|---|---|
| Llama2-7B-chat | -0.068 | 0.473 | 0.979 |
| Zephyr-7B-beta | -0.076 | 1.185 | 1.277 |

Table 2: **(Watermarking, $z$-scores)** Strength of the watermark signal reported as average $z$-score produced when running the detector on the generated answers.

| Watermarking Scheme | None | LeftHash | SelfHash |
|---|---|---|---|
| Llama2-7B-chat | 0.528 | 0.379 | 0.273 |
| Zephyr-7B-beta | 0.528 | 0.237 | 0.222 |

Table 3: **(Watermarking, P-values)** Strength of the watermark signal reported as average P-values produced when running the detector on the generated answers.

## A.7 DATASETS AND MODELS ABLATION

In this section we present the results of an extended evaluation of watermark robustness to machine paraphrasing attacks across a selection of domain specific datasets using the two models utilized in the main experiments `llama` and `vicuna` and a similarly sized but older generative model from the Open Pretrained Transformer family, `opt-6.7b` (Zhang et al., 2022). For cross-reference, the black marker indicates the same standard model and data pair from the evaluations in the main work. The "Github", "Law", "Med", and "Patents" markers refer to the `github`, `free_law`, `pubmed`, and `uspto` subsets of the "The Pile" (Gao et al., 2020) dataset as hosted on the huggingface hub at `huggingface.co/datasets/EleutherAI/pile` by EleutherAI. "Wiki" indicates samples from the training split of Wikitext103 dataset (Merity et al., 2016), also hosted on the huggingface hub at `huggingface.co/datasets/wikitext` as `wikitext-103-raw-v1`.

Generally, looking at Figure 13, we see that the Github subset yields the lowest $z$-scores relative to the number of tokens considered here, most likely a function of the restrictive nature of code syntax. Less flexibility at generation time due to restrictive prompts and domain characteristics results in a lower average "spike-entropy" of the model's next-token distribution. This has a proportional effect on how strongly the watermark can be embedded, due to its adaptivity property (see Kirchenbauer et al. (2023) for a full analysis of the effect of entropy on watermark strength) and implies that under default settings, more text needs to be observed to detect the watermark. Upon manual inspection, the Law subset is also highly formatted containing special characters and whitespacing and this is a potential explanation for the low peak $z$-score (below 8.0) since the model is forced to follow syntax cues in the prompt to maintain a high level of language modelling accuracy. The Med and Patents subsets yield a wider range of $z$-scores across the three models in the 8.0 to 10.0 range, and the C4-en and Wiki datasets produce $z$-scores very similar to the C4-News split considered throughout the main work.

Out of the three models evaluated, the `vicuna` model, which is a supervised instruction-finetuned variant of `llama`, yields the lowest $z$-score for each dataset, suggesting that its output entropy is generally lower than that of the base `llama` model or `opt` in congruence with the fact that we found a higher delta value (4.0) was required to achieve suitably strong starting watermark levels in the human paraphrasing study (details in Appendix A.13).

While in Figure 14 we also present the combination of models and data but with a different metric along the y-axis, we find that the appropriate takeaways are mostly the same as described above. Further, the semantic similarity (P-SP) values for the Github data domain are potentially miscalibrated since this metric was not designed for code similarity estimation. That said, we notice that `vicuna` and `llama` achieve the same $z$-scores across Law, Med and Patents, but show more semantic divergence between the unwatermarked and watermarked outputs for Med and Patents than Law.

In Figure 15 we observe the effect of the different $z$-scores shown in Figure 13 and Figure 14 on the unattacked detection rate. The ROC-AUC is quite a bit lower for Github and Law than the other datasets. Then, in Figure 16 we see that this also translates into correspondingly lower detectabilty after attack by GPT and Dipper. However, we note that for the Med and Law subsets (yellow and green bars), detectabilty post paraphrase attack is quite similar to the C4-News domain (black bars) utilized throughout the main work. This suggests that those findings generalize well at least to language modelling domains with similar levels of syntactic flexibility.

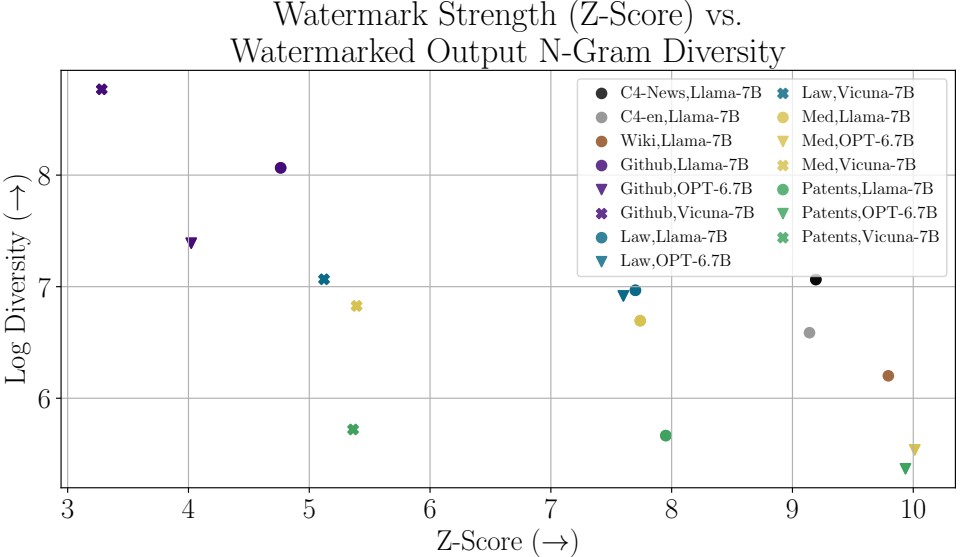

**Figure 13:** The pareto frontier of watermark strength in $z$-score, shown on the $x$-axis, versus text diversity, shown on the $y$-axis. Higher is better for both metrics. The Github subset results in particularly low $z$-scores for all models and the Law subset is also shifted left versus the C4 data splits. Med, Patents, and Wiki yield higher $z$-scores more similar to the C4-News data from the main work.

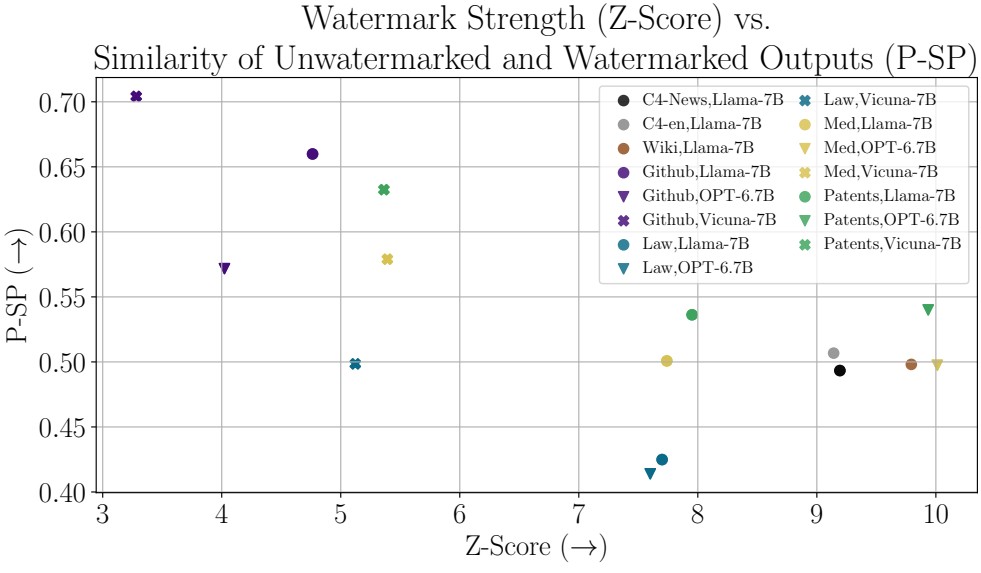

**Figure 14:** The pareto frontier of watermark strength in $z$-score, shown on the $x$-axis, versus similarity between the un-watermarked and watermarked output, shown on the $y$-axis. Higher is better for both metrics. Similar trends to Figure 13.

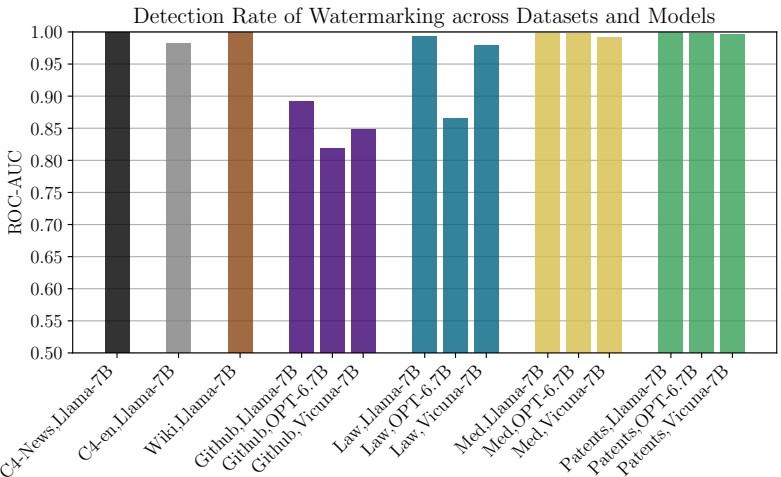

**Figure 15:** Detection rate of watermarking as measured by ROC-AUC for extended selection of datasets and models. The Github and Law subsets producing lower $z$-scores in Figure 13 corresponds to lower ROC-AUC. The `vicuna` model yields the least detectable watermark out of the three models in most cases.

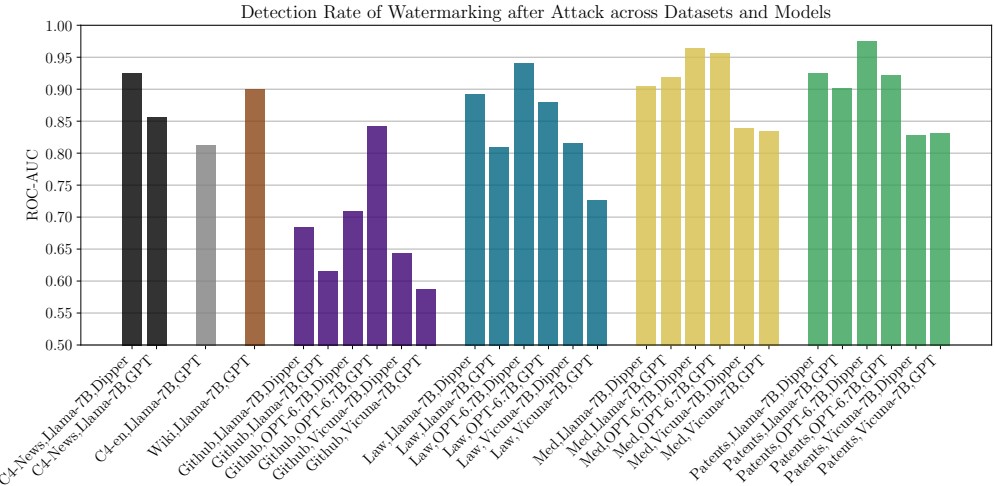

**Figure 16:** Detection rate of watermarking after being attacked using a machine paraphrasing model as measured by ROC-AUC for extended selection of datasets and models. The Github and Law subsets producing lower ROC-AUC in Figure 15 corresponds to lower ROC-AUC after attack as well. While the GPT attack is more successful in reducing the detectability of the watermark in a majority of the cases (around 8/12) a stronger claim is withheld without further ablation of domain specific paraphrase instructions or parameters for the dipper paraphrasing model.

## A.8 GPT ATTACK PROMPT ABLATION

When using `gpt-3.5-turbo` (GPT) to paraphrase the text throughout the experiments on watermarking and detector robustness, we prompt the model with an instruction prompt for the paraphrasing task. In our initial experiments we observed the tendency of GPT to produce a *summary*, i.e. it generated text with good semantic similarity to the input but significantly shorter overall length with this summarization ratio worsening as the original length was increased. We tried to address this issue by specifically adding directives to the prompt to encourage the model to maintain the length of the text. While ultimately we were unable to solve this problem through our basic prompt engineering, we enumerate the prompts we explored in Table 4. We leave the further study of how to optimally prompt general purpose LLMs such as GPT for paraphrasing tasks to future research.

We note that in one sense, this makes the attack using GPT strictly stronger, as some information contained in the original text is left out during summarization.

The prompt we selected to use throughout our experiments was "Prompt 4" as it resulted in a slightly lower ROC-AUC for the standard $z$-score detector (lower indicating a stronger, more successful paraphrase attack) than "Prompt 3" whilst achieving a slightly longer averaged attacked output length than "Prompt 0". Prompts 1 and 2 were included in table to be consistent with a file in the source code though those other prompts were not competitive so they were omitted from the figures.

| Prompt ID | Prompt Text |
|:---:|:---|
| 0 | "paraphrase the following paragraphs:\n" |
| 1 | "paraphrase the following paragraphs and try your best not to use the same bigrams from the original paragraphs\n" |
| 2 | "paraphrase the following paragraphs and try to keep the similar length to the original paragraphs\n" |
| 3 | "You are an expert copy-editor. Please rewrite the following text in your own voice and paraphrase all sentences. \n Ensure that the final output contains the same information as the original text and has roughly the same length. \n Do not leave out any important details when rewriting in your own voice. This is the text: \n" |
| **4** | "As an expert copy-editor, please rewrite the following text in your own voice while ensuring that the final output contains the same information as the original text and has roughly the same length. Please paraphrase all sentences and do not omit any crucial details. Additionally, please take care to provide any relevant information about public figures, organizations, or other entities mentioned in the text to avoid any potential misunderstandings or biases." |

**Table 4:** GPT paraphrase attack prompts. Performance of 0,3,4 shown in Figure 17 and Figure 18 as prompts 1 and 2 were not competitive. Prompt 4 used throughout experiments in main work and Appendix.

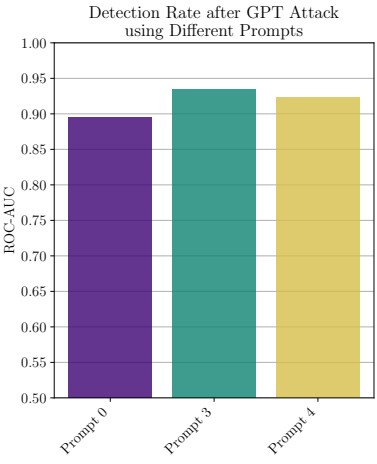

**Figure 17:** Detection rate of watermark after attack by GPT using various prompts.

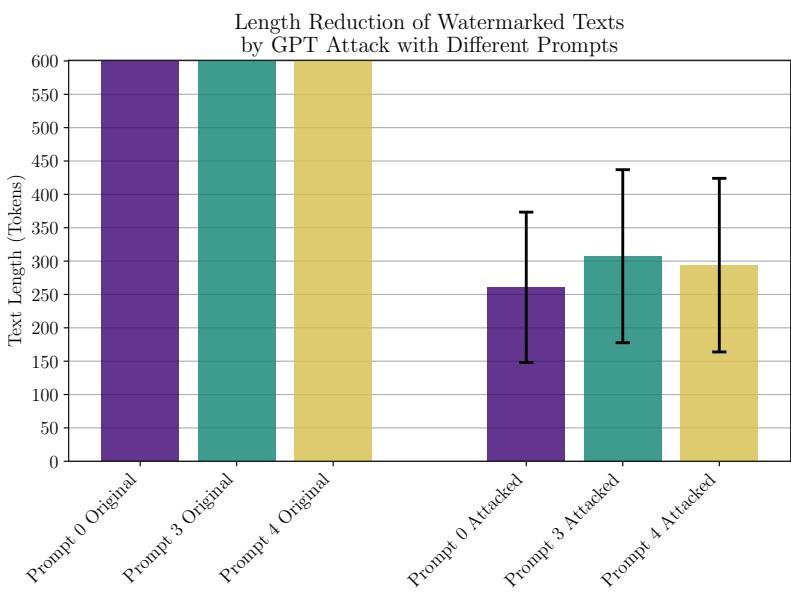

**Figure 18:** Original token lengths (all 600) versus token length after paraphrasing using different prompts for GPT. Standard deviation is visualized for the latter. We choose "Prompt 4" to balance AUC after attack and length after attack, but a wider study of prompting general purpose LLMs for paraphrasing tasks is left to future research.

## A.9 WATERMARK DETECTOR ABLATION

### A.9.1 WINDOW-BASED METHOD

We design a windowed test, called **WinMax** to accurately detect watermarked regions even in long documents. This is an alternative way of formulating a detection hypothesis that can be employed optionally or in conjunction with the original test and requires no modification of the generation scheme. Given a sequence of tokens, we first score the sequence on per-token basis to find the binary vector of hits $s \in \{0, 1\}^T$ to each green list, which we can convert to a partial sum representation $p_k = \sum_{i=1}^{k} s_i$. WinMax searches for the continuous span of tokens that generates that highest $z$-score. More formally, it computes

$$z_{\text{win-max}} = \max_{\substack{i,j, \\ i<j}} \frac{(p_j - p_i) - \gamma(j - i)}{\sqrt{\gamma(1 - \gamma)(j - i)}}. \tag{4}$$

As this test involves multiple hypothesis testing, we calibrate to a fixed false-positive rate based on comparisons with non-watermarked text.

### A.9.2 RUN-LENGTH BASED METHOD

We further investigated a more complex anomaly detector based on run-length differences between watermarked and unwatermarked text (Bradley, 1960). Yet, we found no gains from such a detector over $z$-test and WinMax within the range of settings we consider in this work. We include a brief description of this alternate detection algorithm as a starting point for future research.

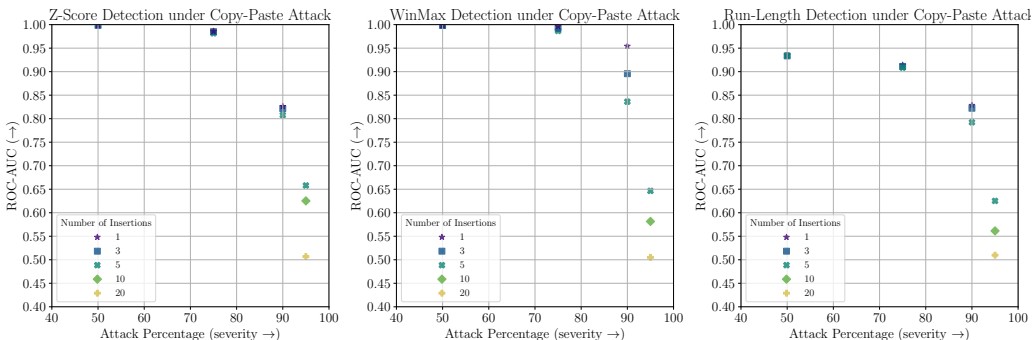

Figure 19: **(Left)** The performance of standard Z-Score watermark detection, **(Center)** WinMax detection, and **(Right)** the Run-Length based detector. WinMax outperforms the standard Z-Score at the 90% attack percentage, and the run-length test failed to show improvement over the standard test at any setting evaluated, however it demonstrates parity at stronger attack levels.

Additionally, we investigate an anomaly detection test based run-lengths as a potential detector of the distribution shift between watermarked and unwatermarked text (Bradley, 1960). Yet, we find no gains from such a detector within the range of attack settings we consider in this work. After describing the basic results, we include a brief description of this alternate detection algorithm as a starting point for future research.

In Figure 19, moving right along the x-axis means that the attack on the watermark becomes more severe (higher "attack percentages") because less of the total text remains watermarked. The marker style denotes how many fragments the remaining watermarked percentage is distributed across in the final text. As an example, the blue square at 90% attack, means 10% total tokens watermarked, split across 3 chunks. The WinMax method shows comparable performance with the standard Z-Score method at all settings except for the 90% attack percentage, where it handily outperforms the standard detector.

For the rightmost plot in Figure 19 showing the run-length detector performance, we visualize the best performing variants of the run length test, where we only count the green runs, we use the "maximum-plus-1" binning strategy, we ignore any run-lengths for which we recorded zero observations, and we use the standard "pearson" chi-squared test statistic to compare expected and

observed frequencies. Additionally, for the more severe 90% and 95% attack levels, the best performing setting was to ignore the length-1 runs bin in the test statistic computation. These details are elaborated at the end of this section.

**Counting the Lengths of "Green Runs"** At detection time, the test originally proposed by Kirchenbauer et al. (2023) initially treats a piece of text as a sequence of green and red tokens, which can be modeled by a Boolean array i.e. sequence $\mathbf{s}$ such that $s_t = 1$ if $s_t \in G_t$ and $s_t = 0$ if not. However, the z-test then immediately reduces this sequence to one value $|\{s_t : s_t \in G_t\}|$, the number of green tokens observed. We hypothesize that this reduces the power of the test in certain scenarios because it does not take into account information regarding the positions of the green (and red) tokens.

To harness this information, one can view the Boolean array as a set of *runs*, where a run is defined by a subsequence of consecutive 1's or 0's. As an example, under the convention that a 1 corresponds to a token being in its greenlist, the sequence $[1, 1, 0, 1, 0, 0, 0, 1]$ contains 2 green runs of length 1, and 1 green run of length 2. It also contains 1 red run of length 1 and 1 red run of length 3.

The example scenario that motivated our exploration of this method was the text mixing setting (copy-paste attack) where sections of watermarked text would be interspersed within surrounding unwatermarked text. In the case of just a few heavily watermarked subsequences, one would expect to observed a few isolated runs of green tokens (consecutive $1's$) that were surprisingly long. This is because we expect to observe few greens ($\gamma T$ in expectation) in unwatermarked text, and many more in heavily watermarked text, and long green runs are themselves caused by a higher overall green token count.

**Hypothesis Test for a "Run-Length Detector"** To formalize this notion of what we'd find "surprising", and thereby derive a new hypothesis test from it, we leverage the fact that for a binary event with a probability of "success" $p$, the number of independent trials $k$ required to realize the first success can be modeled by a geometric distribution, with density given by

$$Pr(\text{"success after k trials"}) = (1-p)^{k-1}p, \text{ for } k = 1, 2, 3... \tag{5}$$

We can treat observing a $0$ or a redlist token as a *success* event, and therefore model the "green runs" as a geometrically distributed random variable with success probability $1 - \gamma$. Armed with this fact, if we treat each run length $k = 1, 2, 3...$ as a category, then for a given piece of unwatermarked text, the expected values of each of these categorical variables can be computed using the geometric distribution density function scaled by the total number of runs observed in the text.

Therefore, we can test for the watermark by testing the following null hypothesis,

$$H_0\text{: The text sequence is generated with no knowledge of the watermarking rule}$$
$$\text{and therefore the "green" run lengths are geometrically distributed.} \tag{6}$$

One standard way to compare an observed set of categorical variable outcomes to an expected set, is to perform a chi-squared test (Cochran, 1952). In this test the sum of squared deviations between the expected and observed frequency (count) for each category are summed. If the statistic is small, then the oberved counts are close to the expected, and we fail to reject the null hypothesis. However, if the statistic is large, then at least one of the observed frequencies is surprising different from the corresponding expected frequency and the collection of categorical variable observations is unlikely to have come from the expected distribution, in which case we can reject the null hypothesis.

Returning to the context of green run lengths, this means that if we observe green run length counts that don't match the expectation given by the geometric distribution, we are confident that this sequence was not generated under the null hypothesis, and rather, was generated under the watermarking scheme. We expect that the most common mechanism for this rejection under watermarking would be observing a surprising number of long runs of green tokens, i.e. surprisingly high frequencies in the tail (larger values of $k$) than expected under the geometric distribution.

**Design Choices**

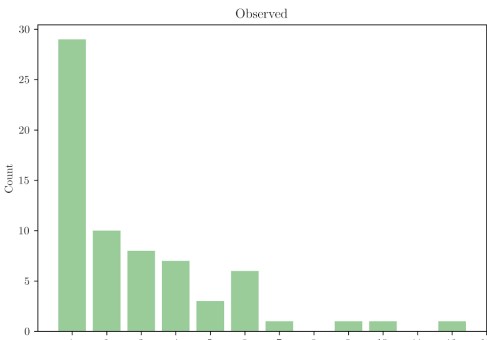 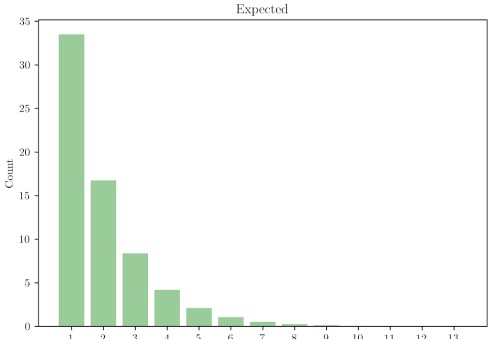

**Figure 20:** An intuition-building example of the run length distributions we seek to compare. The left is the actual, empirical run lengths observed in a *watermarked* sample, and the right is the expected counts of each run length based on the total number of observed runs, dsitributed according to the geometric prior parametrized by $1 - \gamma$ (the null hypothesis). Each bar shows the number of runs of a given length that were observed. The "surprising" observations are the non-zero counts for the length 7,9,10 and 12 bins.

1. When we "count green runs, where red is the success event" a new run begins after each observation of a red token. The sequence $[R, R, R, G, R]$ yields 3 "green runs" of length 1, and then 1 green run of length 2. This is because a red occurs after just a single trial, three times in a row, and then the final red takes two trials to achieve.

2. For a sequence of length $T$ one could observe any possible run length $k \in \{1...T\}$, and we can compute an expected frequency for all $k$ based on the success probability $1 - \gamma$. However, it is standard practice to bin the tail of unobserved $k$ values to create a new event "run lengths longer than k" and the probability mass for those values is summed before computing the expected number of outcomes for that tail category. In Figure 20, the max shown (13), is 0, but the maximum actually observed was 12 and this addition of an extra bin represents the tail of runs longer than the max observed.

3. For any run lengths $k$ between 1 and the largest observed run length value $k_{max}$ it is standard practice to ignore these categories when computing the test statistic if there were zero observed occurrences. This is based on the assumption that the zero observation is likely spurious consequence of a small sample size (of runs). In Figure 20, there are two bins, 8 and 11, that also are zero, which can be ignored in the test statistic computation.

4. We consider the standard "pearson" formulation of the chi-squared test, as well as the "g-test" and "cressie-reed" variants based on likelihood ratios rather than squared deviations.

5. In order to isolate the rejection scenario we expect under watermarking, we experiment with ignoring the small categories $k$ in the test statistic computation. The intuition is that these short run lengths could dominate the statistic value in an undesirable way when there are just a small handful of surprisingly long runs in the tail of the observed distribution. Considering the example in Figure 20, since the close counts of observed and expected length-1 runs potentially of little interest with respect to the null hypothesis reject case expected for the watermark, we can choose to ignore the leading bin in the test computation.

## A.10 DETECTABILITY OF WATERMARKS AFTER MACHINE PARAPHRASING: TPR @ T

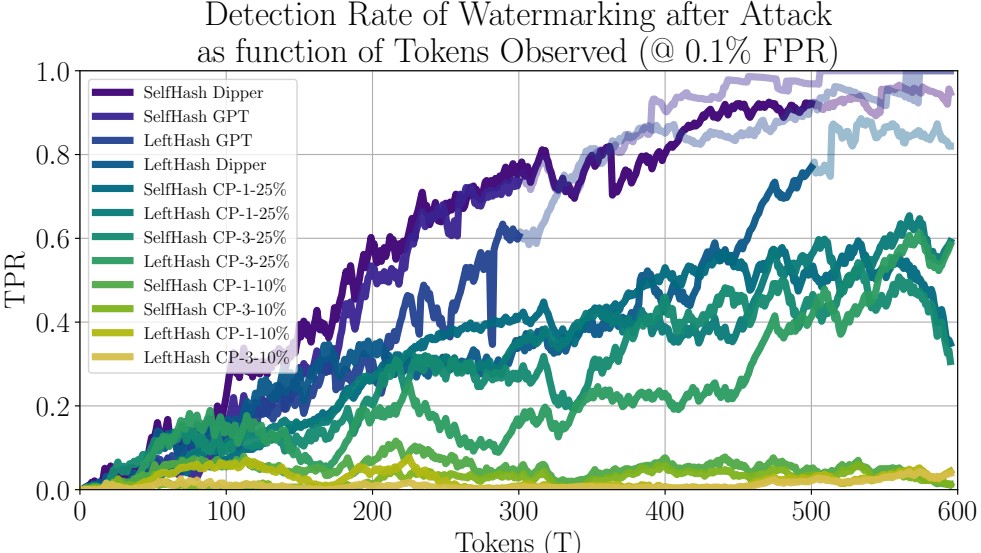

**Figure 21:** In this variant of a plot from the main body, we visualize the growth characteristics of watermark detection as quantified by True Positive Rate at low False Positive Rate (0.1%) after attack by a selection of machine-based attacks. As in the main body charts, we make the curves translucent starting at the mean sequence length of the attacked set after generation to indicate that these measurements are based on increasing fewer samples after that point as T continues to grow and are therefore more uncertain. For the Dipper attack this is $\sim 500$ and for the GPT attack this is $\sim 300$. Due to the synthetic nature of the Copy-Paste attack, after attack, those sequences are still full length i.e. $600 \pm 25$.

## A.11 Experimental Methodology Details

### A.11.1 GPT Paraphrase Attack

We utilize the prompt chosen through the ablation in Appendix A.8 and query the model using a sampling temperature of $0.7$ and a max token limit of $1000$.

### A.11.2 Dipper Paraphrase Attack

We utilize the Dipper paraphrase model proposed by Krishna et al. (2023) and released at their github. Since this model smoothly trades off semantic similarity between the paraphrased and original text starting from few discernible changes at one end to almost wholly unrelated at the other extreme of its `lex` and `div` parameters, we choose one of the moderate strength settings to run the paraphrasing attacks with across all experiments `lex=40,div=40`. This still represents a significant attack, but maintains high paraphrase quality/similarity. We of course emphasize that the setting used is the same for all the watermarking and baseline detector methods to fairly compare them.

### A.11.3 Baseline Method Details and Hyperparameters

We lightly adapt the codebases provided by the authors to interface with our generation, attack, and evaluation pipeline for both methods and use default parameters found in their code unless otherwise specified.

**Retrieval:** The retrieval system of Krishna et al. (2023) requires the creation and maintenance of a comprehensive database of all sequences generated previously by the language model, so that paraphrased generations can be mapped to their original, uncorrupted source. In our experiments, we adopt the retrieval method as described by (Krishna et al., 2023), and utilize the BM25 search method as it performed better in their evaluation. A key detail concerning this method is how we construct copy-paste examples to evaluate it on. The "unattacked" text is the output of the language model without any modification and this is what is loaded into the retrieval database as the "generation history" of the model. The copy-paste attacked version of the text is created by inserting a sub-string from that text into the other piece of machine-generated text readily available for this prompt, the *watermarked generation*.

**RADAR:** To strengthen the performance of the learned classifier approach in the context of text corruption and paraphrasing, Hu et al. (2023) proposes RADAR, a framework for training a robust AI-text detector using adversarial learning. Their method is inspired by techniques such as generative adversarial network (GAN) training which poses learning as a two-player game. RADAR comprises a paraphraser and a detector model where the paraphraser's goal is to generate realistic content that can evade AI-text detection, while the detector's goal is to enhance AI-text detectability. We use the only publicly available RADAR-Vicuna-7B model, where the detector model has been robustly optimized to detect Vicuna generations.

**DetectGPT:** DetectGPT (Mitchell et al., 2023) is a zero-shot post-hoc method for detecting machine-generated texts employing a curvature-based criterion that compares the log probabilities between a candidate passage and perturbations of it. The intuition behind DetectGPT is that machine-generated texts tend to dominate the negative curvature regions of an LM's log probability curve. We use the official implementation of DetectGPT and follow their default setting by using `T5-3B` (Raffel et al., 2020) as the mask-filling model for generating perturbations. We adopt the strongest setting of DetectGPT by generating 100 perturbations for each test sample and using the normalized perturbation discrepancy as the criterion ("$z$" in their notation). Further details regarding the evaluation of the method are included in the appendix, but the key experimental detail is that the positive examples for detection are derived from the unwatermarked model outputs generated from each prompt, and as with watermarking detection, the human gold completions are the negative examples.

We remark that there are some unknowns about how well the DetectGPT paradigm works when different models are used as the detection target, especially when under attack. In this work we primarily utilize `llama` as the base model to be detected, and the relationship between the relative sizes and qualities of the base model, the perturbation model (a 3B parameter version of T5), and

the machine paraphraser (Dipper or `gpt-3.5-turbo`), could be quite subtle and produce counter-intuitive detection performance outcomes. We leave the study of post-hoc detectors with improved robustness characteristics to future research.

### A.11.4 DEFINING "POSITIVE" AND "NEGATIVE" DETECTION TEST SAMPLES

**"Negative" Samples**  In all experiments, either using watermarking as the detection method or the two baseline approaches, the "negative" samples at test time that should be classified as "not watermarked" or "not machine-generated" respectively, are human-written text i.e. the gold completions/suffixes extracted from the input data as prompts are created.

**"Positive" Samples for Watermarking Detection**  For the watermarking experiments, in the unattacked setting, the "positives" are the watermarked model's generations. To produce the paraphrase-attacked versions, the watermarked generation is fed to the paraphrasing model (GPT or Dipper) or rewritten by humans. To construct the copy-paste attacked versions for watermarking, sets of watermarked tokens are inserted into a surrounding context sequence of *human-written* tokens (i.e. the gold completions to the prompt). While this means that the negative examples and surrounding context examples for the copy-paste watermarking experiments are correlated, this does not give the watermarking method any unfair advantage at detection time. If a specific sequence of human-written text happens to produce an abnormally high $z$-score, which *could* artificially inflate the $z$-score of the copy-paste attacked example making it an easier detection, then it will simultaneously increase the chance of a false positive on that sequence for precisely the same reason. Since both examples are always tested, this effect should be balanced out.

**"Positive" Samples for Alternate Detection Approaches**  For the Retrieval based and Detect-GPT baseline approaches, the "positives" are the unwatermarked model's generations as this is the distribution that the approach is designed to detect. For the Retrieval method, this means that the retrieval database (index) is loaded up with the set of unattacked, unwatermarked generations so that, if those same sequences are queried at test time, then the retrieval performance (as measured by TPR) should be perfect. To construct paraphrase attacked examples in this setting, the unwatermarked generations are fed to the paraphrasing model (GPT or Dipper) causing them to diverge from the exact sequences present in the database, and the exact model distribution that DetectGPT is testing for.

**Copy-Paste Attacked Samples for Alternate Detection Approaches**  As stated above, for copy-paste attacks in the the watermark detection evaluation, we insert spans of watermarked tokens into surrounding context of human-written tokens. Since watermarking effectively views all text as boolean arrays "green" and "red" tokens regardless of the textual content itself, human-written text is the most "negative" type of context we can create to make the embedded watermarked chunk harder to detect.

However, since the human-written completions are already used as the negative examples, and the baseline detection methods rely much more on the syntactic and semantic similarities between the examples, to reduce the error correlations between the negative and positives for Retrieval detection and DetectGPT, we instead insert the small unwatermarked chunks (parts to be detected) into a surrounding context of *watermarked* text. While we realize this choice might seem a bit strange, and also admit that it is mostly an implementation pipeline convenience rather than a perfectly optimal choice, we do not believe this causes any unfair biasing of the detection performance estimates for the following reasons.

For a fair copy-paste example with respect to the particular detection approach being evaluated, we desire "negative" looking context tokens to surround the "positive" looking chunks to be detected. From a semantic similarity perspective, the watermarked generations used as a source of context tokens, are quite relevant/similar to the unwatermarked subsequences being inserted because they were generated following the same prompt. For Retrieval, this means that there is actually a generous/favorable level of semantic similarity between the context tokens (meant to make the copy-paste attacked sample look negative) and the unwatermarked outputs stored in the retrieval database. We believe this is a reasonably fair and realistic setting since the copy-and-pasting attacker we are simulating would likely replace removed sections of the text to be detected with semantically similar

and relevant text (as opposed to random tokens). However, overall, this is likely an overly generous setting for these methods.

For the RADAR method, we construct the copy-paste examples in the same manner as for the watermarking methods. We paste sets of un-watermarked tokens into a surrounding context sequence of human-written text. This has the same correlation caveat as described above for the watermarking case, but we felt it necessary to use this evaluation setting given the fact that the RADAR method was demonstrated to exhibit strong transfer properties between models. Since the watermarked completions are in fact quite similar to the un-watermarked pairs, this would present copy-paste examples that were effectively almost not attacked at all from the perspective of the RADAR model.

**Potential Confounding Factors in Copy-Paste**   We believe that this setup is fair for the Detect-GPT method, however, we realize that both the perturbation procedure and the likelihood estimation are probably influenced by certain discontinuities introduced in copy-pasted examples as no algorithm or post-processing step is used to smooth out the interface region between the positive and negative spans. That said, since we find that Retrieval performs quite well under the copy-paste attack and that for DetectGPT the copy-paste examples produce somewhat unremarkable behavior as visualized in the main work and in Appendix A.12 (versus the GPT and Dipper results), we believe these methodological choices did not significantly influence the results in an unfair way. That said, we believe there are still open questions in developing suites of different attack strategies ranging the gamut from black-box model-based paraphrasing to synthetic text mixing that test a wider range of possible attack and corruption scenarios that could be encountered in-the-wild.

### A.11.5   BASELINE DETECTION METHOD PERFORMANCE AS A FUNCTION OF TOKENS OBSERVED

While the watermark detection score ($z$-score) is readily computable in a cumulative manner along the sequence length dimension, and thus evaluation at T is simple for that method, for Retrieval and DetectGPT, the sequences to be tested must first be chunked into prefixes and then those sets of prefixes evaluated individually. For Retrieval detection, we make the choice that the sequences loaded into the database should be the *full length* unwatermarked output, even though we are querying with prefixes of those outputs to develop the series of measurements at T. This reflects what we believe to be a realistic setting where the model generated the full output at some time in the past, and at test time is being queried with a prefix/fragment of the original generation. Additionally, storing all prefixes of some given size for all generations is not a realistic or scalable choice in our opinion.

For DetectGPT, the adaptation for detection at T is easier in one aspect because it is simply implemented by testing a block of prefixes of the original unwatermarked (and then attacked) generations. However, the computational cost of producing just a single series of measurements at a range of T values becomes prohibitively expensive for a single attack setting. This is because for each block of prefixes, say the leading 100 tokens from a set of N sequences that originally had length 600, the runtime cost is roughly the same as testing the full length outputs. Thus the overall runtime of testing all prefixes, or T values, for a given set of sequences is multiplied by the number of prefixes, or length/stride. In contrast, there is effectively no multiplier for watermarking, and it is still relatively cheap for Retrieval since the retrieval method itself is much cheaper (at least for a small database). This is the reason for the limited number of datapoints shown in the comparisons for DetectGPT as a function of observed tokens.

As final remarks on the somewhat surprising performance of the method, we note that the generations are tested without the original prompts included in the input. We assume this is the setting of the original work, since having access to the prompt at test time would be unrealistic, but beyond this detail, we are unsure as to why evaluating the method using the `llama` model as the base model to be detected, performs as poorly as it does. We leave more comprehensive robustness evaluation of the DetectGPT method to future work by its creators and other researchers but hypothesize that the relationship between the base model, the perturbation model, and the paraphrasing model, could be more nuanced and require more careful tuning of hyperparameters or other aspects of the detection method.

### A.12 DETECTION METHOD COMPARISON: DETECTOR SCORES @ T

In this section we present a "mechanistic" explanation for the detectability as a function of text length (AUC at T) results reported in the main body of the work. In particular, we motivate the perusal of this section by remarking that in order to perform well, a score-based binary detector (like all the methods considered here) must maintain a gap between the scores assigned to "negative" or genuine human written samples, and the "positive" or machine generated samples. The left chart in each pair are the scores assigned to negative examples by the given method, and the right chart shows the scores for positive examples. We make note of the fact that the standard Z-Score watermarking produces an *extremely* wide gap between corresponding curves in the left and right charts in Figure 22, driven in large part by the lack of growth in the negative scores. This key behavior of watermarking provides for both detectability and a low FPR.

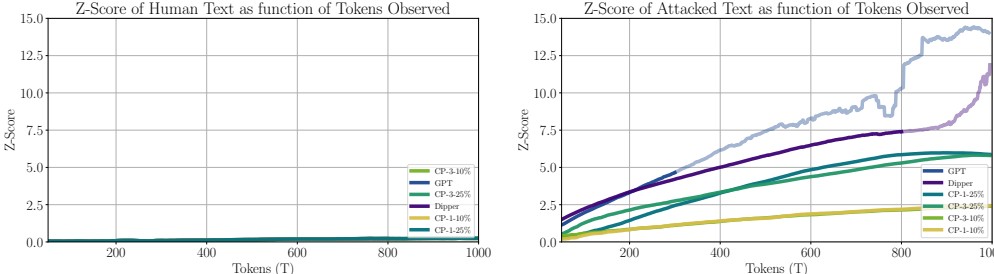

**Figure 22:** The deterction scores yielded by the watermarking detection z-score method. In stark contrast to the trends in Figure 24 and Figure 25, we see that for all $T$, the $z$-score of human text remains very low (**Left**) whilst the $z$-scores for the attacked watermarked texts continue to grow steadily (**Right**) demonstrating the favorable token sample complexity characteristics of watermarking detection. As in preceding figures we turn the curves translucent after their mean sequence length value to indicate increased uncertainty.

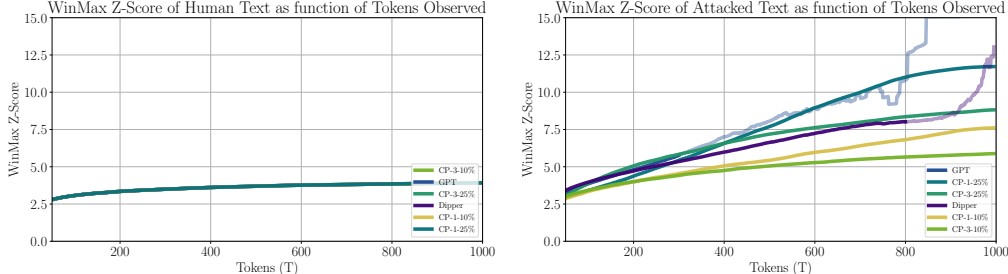

**Figure 23:** The detection scores yielded by the watermarking detection WinMax z-score method. Compared to Figure 22 we see that the likelihood of a False Positive at smaller values of T is higher under WinMax than the basic $z$-score detection test as the separation between the scores for human text (**Left**) and attacked watermarked text (**Right**) is not as large. However, empirically, this tradeoff enables improved detection under the strongest copy-paste attacks as shown in Figure 5 and Figure 6. As in preceding figures we turn the curves translucent after their mean sequence length value to indicate increased uncertainty.

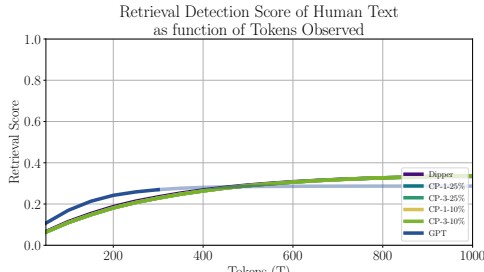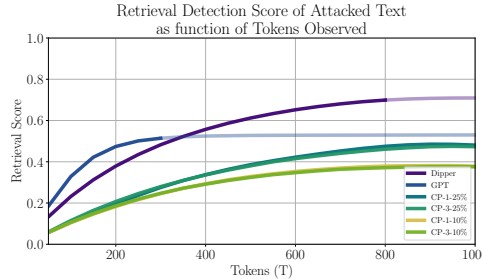

**Figure 24:** The similarity scores yielded by the Retrieval detection method using the BM25 search index. (**Left**) The corresponding scores for "negative" human-written test samples are lower than (**Right**) the scores for the "postive" attacked samples for all values of $T$, which is desired/required for proper detection and mechanistically explains the favorable detection performance shown in other figures. However, we note that the gap is not as large for the copy-paste attack as it is for GPT and Dipper. As in preceding figures we turn the curves translucent after their mean sequence length value to indicate increased uncertainty.

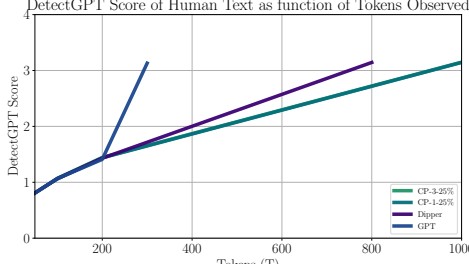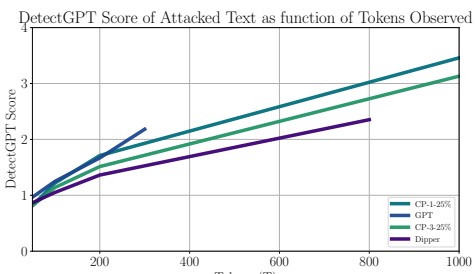

**Figure 25:** The similarity scores yielded by the DetectGPT method. (**Left**) While the corresponding scores for "negative" human-written test samples start out lower than (**Right**) the scores for the "postive" attacked samples at small values of $T$, which is desired/required for proper detection, this ordering becomes reversed for the GPT and Dipper paraphrase attacks at larger T which helps explain the unfavorable/"inverted"-looking detection performance curves out at 300 and 800 tokens for those methods shown in other figures.

### A.13   HUMAN STUDY DETAILS AND PREFERENCE EVALUATION

#### A.13.1   INDIVIDUAL HUMAN PARAPHRASER PERFORMANCES

Examining the individual performances in Figure 26, the only real exception in this study was human writer 13, who is a strong paraphraser and simultaneously did not submit enough text to be detected (human 24 only partially completed the tasks and was omitted from gift-card consideration). More text from this writer would be required to guarantee detection. On the other extreme, human writers 22 and 15 apparently paraphrased the text with a strategy that did not substantially affect the watermark, and are reliably detected after only about 250 tokens, or 200 words.

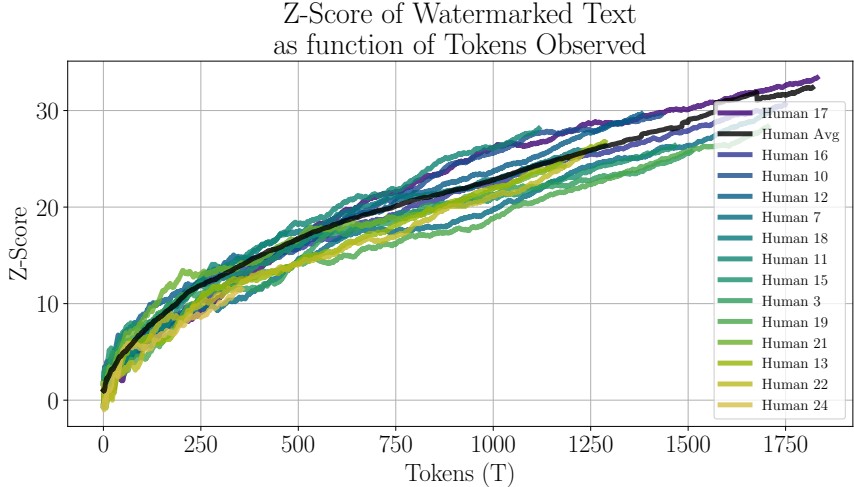

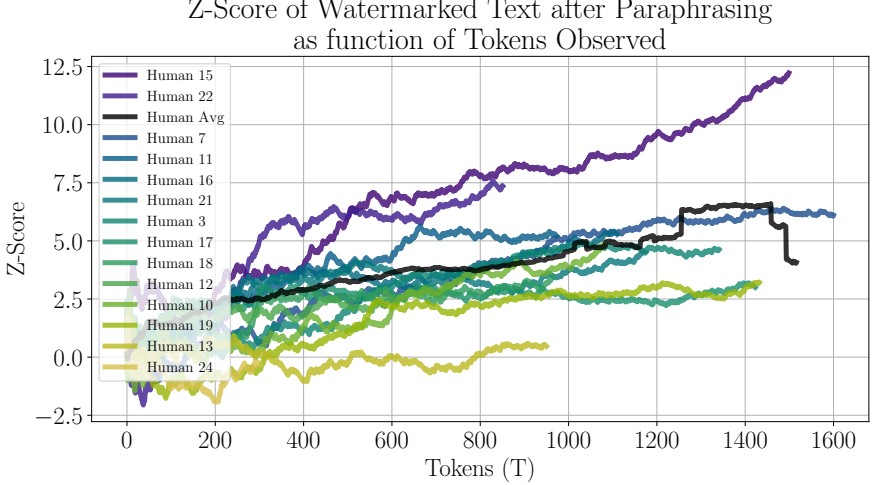

**Figure 26:** $z$-score as a function of $T$ over all text passages given to each writer, separated per writer. **(Top)** Original scores for the combined text given to each human writer. **(Bottom)** Average scores per writer after paraphrasing. We find that almost all human writers are detected with exceeding certainty, after about 800 tokens, i.e. about 500 words (a $z$-score of 4 implies a FPR of 3.2e−5). Note that despite their strong paraphrasing capability (as shown in the left plot of Figure 4), Human 24 paraphrased too few examples for a fair competitive comparison with the other annotators, but they are still included as part of the averages.

#### A.13.2   QUALITY/PREFERENCE EVALUATION

To give a second perspective on the quality impact of watermarking on generated texts, as part of our Human study (Table 5), we ask annotators to rate which of two potential responses to a prompt

they generally prefer. And we find that for a *very* strong watermark ($\gamma = 0.25, \delta = 4.0$) humans only prefer the unwatermarked text over the watermarked text a weak majority the time.

| Total Ratings | Unwatermarked Answer | Watermarked Answer | Unwatermarked Preferred |
|---|---|---|---|
| 177 | 109 | 68 | 61.58% |

**Table 5:** Outcome of human preference study. We report the frequency human evaluators preferred the unwatermarked generation output over watermarked output.

### A.13.3 DATA GENERATION AND SELECTION PARAMETERS

We utilize the Long Form Question Answering (LFQA) style dataset curated by Krishna et al. (2023) available via the Google Drive link provided at `github.com/martiansideofthemoon/ai-detection-paraphrases`. We generate machine responses to the questions using the `vicuna` model under a ($\gamma = 0.25, \delta = 4.0$) watermark prepended with the following prompt:

```
''Answer the following question in 200-300 words.  Explain it like I'm five.\n\n''
```

To select the small subset of examples presented to annotators in the paraphrasing study we filter the original 2758 questions to a subset of 60 by selecting examples such that the watermarked model response was **1)** longer than 200 tokens, **2)** had a z-score of $> 9.0$, **3)** had a P-SP similarity score between the gold human response and the watermarked response of $> 0.6$, and **4)** had a 4-gram repetition rate $< 0.11$. In particular we enforce 1) and 2) to make sure that these examples were of adequate length and heavily watermarked to start out with, in order to develop a significant result based on the final z-scores achieved through paraphrasing. Considering weakly watermarked examples with low starting z-scores, would make for uninformative samples since there would be little watermark to scrub away in the first place. Constraints 3) and 4) were enforced simply to raise the quality of the machine responses as these questions are quite challenging to answer well even for the stronger instruction-tuned `vicuna` model utilized.

For the preference study we filter the original set from 2758 to 205 by enforcing that **1)** token lengths of both the unwatermarked and watermarked outputs were $> 200$ and differed in length by no more than 50 tokens **2)** that the watermarked text z-scores were $> 4.0$. The second constraint was chosen to increase the likelihood of the unwatermaked and watermarked texts being perceived as different based on the significant watermark (most examples had a much higher z-score than that lower limit) and the first constraint was chosen to remove the spurious differences annotators might perceive due to length differences which are not necessarily indicative of quality.

### A.13.4 HUMAN STUDY ANNOTATOR INTERFACE

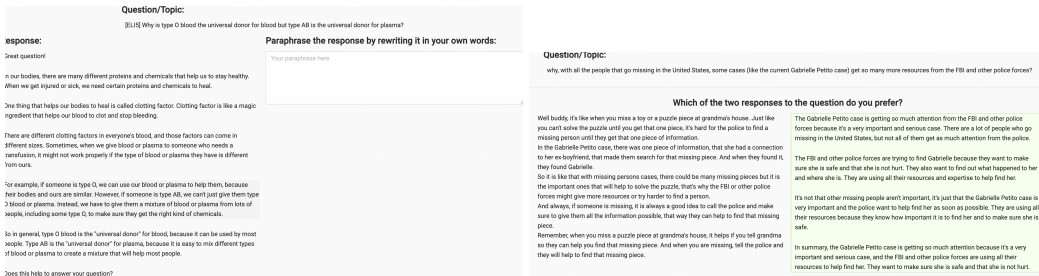

**Figure 27:** The LabelStudio interface designed for the human paraphrasing and preference studies. Left: Interface for paraphrase study. Right: Interface for human preference study.

### A.13.5 ANNOTATION PLATFORM AND TASK INSTRUCTIONS

We utilize the open source data annotation platform LabelStudio (Tkachenko et al., 2020-2022) to conduct the human study and a screenshot of the interfaces constructed for each of the two tasks are provided in the main work.

Here, we additionally show the instructions that were given to annotators for both of the human study tasks in Figure 28 and Figure 29.

## "Paraphrase Text"

### Description

Paraphrase an AI generated response to a question from Reddit's r/explainlikeimfive (ELI5) forum.

### Instructions

A question or topic statement is shown at the top of the screen. On the left side of the screen you will see a response to the question. The response was generated by an AI language model. The response is "watermarked," meaning it contains invisible patterns that can be used to determine that the response was written by an AI and not a person. Read the AI-generated response on the left half of the screen, and in the text box on the right side of the screen, re-write the response in your own words, whilst preserving the meaning and length of the text. Your goal is to change the text so much that the watermark is no longer detectable.

When you are finished, click the "submit" button to save your re-written text and move on to the next task.

### Requirements:

1. **Paraphrase quality/similarity** - A paraphrase should convey roughly the same information as the original text, to roughly the same level of detail.

2. **Time limit** - Try to spend no more than **10 minutes** on any individual paraphrasing task. The annotation software tracks the time you spend on each task, but it will not

   explicitly enforce the time limit by kicking you off. Please do the tasks in a single sitting.

3. **No automated paraphrasing tools** - Do not use any AI tools that write text for you (e.g., ChatGPT, Grammarly), and do not copy/paste text from any external source. However, you may look things up online, refer to a dictionary or thesauruses, and use a spell checker if such a tool is enabled in your browser window.

**Figure 28:** Annotator instruction sheet for the human paraphrasing task.

## "Compare Answers"

### Description

Select a preferred response to questions from Reddit's r/explainlikeimfive (ELI5) forum.

### Instructions

At the top of the screen you will see a question or topic statement. Beneath it there will be two different responses to the question, one on the left and one on the right. Choose the best response of the two by clicking on the left or right text box. Then click the "submit" button on the bottom right to save your selection and move on to the next task.

### Requirements:

1. **Time limit** - Please spend at most **5 minutes** on each individual response pair. If necessary, briefly consult the internet to clarify the meaning of words or check the correctness of statements.

**Figure 29:** Annotator instruction sheet for the preference evaluation task.

### A.13.6 DEMOGRAPHIC AND COMPENSATION DETAILS

We recruited graduate students from a computer science department to do the paraphrase and preference evaluation tasks. 14 annotators worked on paraphrases and 9 additional annotators worked on preference ratings. One out of the 14 paraphrase annotators did not complete enough samples and so is removed from some of the evaluations in the main work. The goal was a maximum diversity of annotated samples and so each of the original watermarked texts was paraphrased by a single annotator, and roughly $150/177$ preference rating examples were for unique questions.

The group comprised both native english speakers and english-as-a-second-language speakers. We prompted volunteers to self pre-select based on a description of the tasks to be performed, emphasizing that the ability to write high quality paraphrases of a few paragraphs in length was required. Admission to the relevant university requires a high level of English language reading and writing competency as assessed by required standardized testing before admission.

All annotation tasks were performed in one 1.5 hour session and all annotators were compensated with free dinner and drinks. As additional incentive to ensure that the paraphrase task was completed in a "motivated" manner to try and approximate the real incentives of a paraphrase attacker, we informed participants that the three most successful attackers (their paraphrases achieved lowest final detection scores) would be awarded a $100 gift card. An additional gift card was randomly awarded to annotators who only performed the preference comparison task as this task was less goal oriented and thus there was not direct way to quantify success or rank annotator performance.

### A.13.7 INSTITUTIONAL REVIEW BOARD "EXEMPT" STATUS

In preparation for conducting the human paraphrasing and preference evaluation study components of the research, a "Human Subjects Research Determination" form was filed with the relevant Institutional Review Board. Before any portion of the human study was conducted, a determination letter was received communicating the status of "Exempt" for the project proposal, i.e. "Not Human Subjects Research".

## A.14    CODE DETAILS AND RELEASE STATEMENT

We extend the implementation of watermarking developed by Kirchenbauer et al. (2023). For the datasets and models evaluated in this work, we heavily utilize the huggingface `datasets` library and huggingface `transformers` modelling framework. We retrieve pretrained model weights and dataset files from the huggingface hub, with the exception of the weights for the `llama` base model.

The `llama` 7B parameter model weights were retrieved with permission using a presigned URL received via email from the "LLAMA Release Team" to be used for research purposes only in accordance with the license. These weights were then converted to the huggingface format before use. To construct the `Vicuna` model, we retrieved the `lmsys/vicuna-7b-delta-v1.1` weights following instructions at `github.com/lm-sys/FastChat` and merged them with the base `llama` 7B weights.

## A.15    HARDWARE AND COMPUTE DETAILS

The experiments performed in the study were all inference-based and therefore could be run on a single Nvidia RTXA4/5/6000 GPU. The 7B parameter models were run in `float16` during generation of watermarked and unwatermarked responses, but the Dipper model was run in full precision in accordance with author recommendation, and output quality issues observed under the `float16` setting. Additionally, the Dipper model and DetectGPT model were both run on A6000 cards due to the memory footprint required by their larger parameter counts. Generation stages, where unwatermarked and watermarked outputs were sampled, took less than 12 hours. Attack stages using both the GPT (OpenAI API) model and the Dipper model took around 4-6 hours. Evaluation stages where watermarking detection at only the max $T$ value was performed took minutes, and when ROC-AUC's at all $T$ values and all text quality metrics were computed, took less than 2 hours. DetectGPT evaluation took well over 12 hours for a single set ($\sim 500$ samples) of generations with longer token lengths such as $T = 1000$.

