# OpenReview forum: "On the Reliability of Watermarks for Large Language Models"
_ICLR.cc/2024/Conference — ICLR 2024 poster_

### Official Review · Reviewer_jTXD · 2023-10-27

**Soundness:** 3 good
**Presentation:** 3 good
**Contribution:** 3 good
**Rating:** 6
**Confidence:** 3

**Summary:**

The paper investigates the effectiveness of watermarking as a means to identify machine-generated text in realistic scenarios, particularly when dealing with potential attacks, such as paraphrasing, copy-paste, or human rewriting. The primary objective is to evaluate whether watermarks can reliably detect machine-generated text when it's subjected to various forms of manipulation, thus contributing to the broader discourse on mitigating potential harms caused by generative language models.

**Strengths:**

Comprehensive Evaluation: The paper conducts a thorough and comprehensive evaluation of watermarking, considering various real-world attack scenarios, including paraphrasing, copy-paste, and human rewriting. This multifaceted approach provides valuable insights into the strengths and limitations of watermarking in practical settings.

Comparison to Alternative Methods: The paper not only focuses on watermarking but also compares it to alternative detection approaches, including post-hoc detectors like DetectGPT and retrieval-based systems. This comparative analysis enhances the paper's contribution by showcasing the relative merits of watermarking.

**Weaknesses:**

Lack of Theoretical Background: The paper does not delve deeply into the theoretical aspects of watermarking, which could be crucial in understanding the underlying principles and potential vulnerabilities. A more robust theoretical foundation could enhance the paper's overall quality.

Inherent Model Bias: The paper uses a specific language model (llama) for its experiments. While this model is justified and used for practical reasons, the results might not be universally applicable to all language models, which could limit the generalizability of the findings.

Limited Discussion of Social Implications: Given the paper's focus on mitigating potential harms from generative language models, it would be beneficial to include a more extensive discussion of the social implications of watermarking and other detection methods. This could provide a more holistic perspective on the topic.

**Questions:**

1. Can watermarking be further improved to make it more resilient to copy-paste attacks, given the significant performance drop observed under this scenario?

2. How does the performance of watermarking compare to other detection methods in detecting machine-generated text within shorter sequences or fragments?

3. Could watermarking be used in conjunction with other detection methods to enhance overall detection reliability, particularly in complex and adversarial settings?

---

> ### Author Response · Authors · 2023-11-17
> **Response to Reviewer jTXD**
>
> On the desire for more theoretical background on the topic, we rely on the foundation established in Kirchenbauer 2023, where the formulation of the watermark we evaluate is established and analytic results are presented. Over the past year, there have been many other works that develop or analyze watermarking strategies from various theoretical perspectives, especially concerning cryptographic and security properties (Christ 2023, Fairoze 2023). However, not all of these works present a realistic empirical evaluation or evaluate the robustness of proposed watermarks to natural corruptions like paraphrasing and text mixing, and even fewer evaluate watermarks alongside other detection strategies. Thus we embarked on this work intending to help fill that particular gap in the literature.
>
> On model bias, we chose to conduct our experiments on the most capable open source foundation model available (at the time of our human study) within our computational budget. We are unable to consider hosted API models because they don't expose the logit distribution necessary to apply the watermark to them. While we only consider a small group, we do report results with a previous generation model, OPT, and an instruction finetuned model variant, Vicuna, in Appendix A.7.
>
> On societal impacts of watermarking, we have penned the following discussion to be included in the updated draft:
>
>
> > While our work focuses on technical aspects of the reliability of machine-generated text detection, these findings have further societal implications. A strong finding that our study supports is that the problem of detection is simply made _tractable_ through watermarking. Compared to other approaches, we find watermarking to be most reliable in everyday scenarios, where text is taken from a generative model, modified by human  writers or other models and inserted into spans of human-written text. *This suggests that watermarking may be a promising strategy to reduce harm arising from the use of large language models*.
>
> > In pursuit of harm reduction through technology like watermarking, it is crucial to consider several precise details. First, from previous studies, i.e. Kirchenbauer 2023, we know that watermarks can be broken by sufficiently motivated attackers, e.g. using generative attacks. Nevertheless, our results among others show that watermarks can be used to reliably track machine-generated text as it distributed over the internet in non-adversarial scenarios. From this perspective, watermarking technology is flips the problem on its head: whereas before, machine-generated texts were mostly undetectable by default, with advanced watermarks, active effort is required to make the text undetectable. Second, while technologies like watermarking will not be _perfect_, to be deployable in the real world, detection strategies must guarantee low false-positive rates, which have to be weighed against detection sensitivity, as every false positive can be extremely harmful. Other approaches like some of the post-hoc detectors we study cannot consistently provide low false positive rates in all domains, and thus can fail, for example when classifying text from non-native speakers. Third, whether harm is reduced is scenario-dependent. For example, in classroom settings, a false accusation is serious, whereas even positive detection results have only minuscule benefits to the learning experience of students. On the other hand, solutions that detect fake news and spam on social media platforms that might include watermarks as one additional feature in a larger system, might trade relaxed error tolerances for clearer expectations through terms of service that communicate both its utility and potential limitations.
>
> We're open to further feedback or specific topics that you feel we should add.

---

> > ### Comment · Reviewer_jTXD · 2023-11-17
> > **Thanks for the feedback**
> >
> > I would like to express my appreciation for the efforts made by the authors in addressing the comments and incorporating improvements into the manuscript. It is admirable to witness the dedication to enhancing the paper despite the initial concerns.
> >
> > However, some of my concerns still persist to some extent. Firstly, I would appreciate it if the authors could delve deeper into providing theoretical insights. Specifically, it would be beneficial for the readers to understand the key takeaways and insights that they can apply to their adjacent work. Clarification on what sets this work apart and how the proposed approach stands out would be valuable. The current response appears somewhat high-level, and additional detail in this regard would be highly beneficial.
> >
> > Secondly, I am eager to read the revised version of the paper. Could you kindly confirm whether the new version has been uploaded? I am particularly interested in reviewing the highlighted changes to gain a clearer understanding of the modifications made.
> >
> > I look forward to your response and appreciate your continued efforts to enhance the manuscript.

---

> ### Author Response · Authors · 2023-11-17
> **Response to Reviewer jTXD (continued)**
>
> > How can watermarking be further improved to make it more resilient to copy-paste attacks, given the significant performance drop observed under this scenario?
>
> We find the window detection strategy (WinMax) to be an effective improvement over the original scheme. These results are exemplified by the the CP-1-10% and CP-3-10% sub charts in Figures 5 and 6. The WinMax scheme retains significantly more detection performance than the standard Z-Score method at this extreme setting.
>
> Regarding the search for even further improvements in this setting, we include a small investigation into another alternate detection test  based on the concept of "run-lengths" in Appendix A.9.2, but it did not yield any improvements in our experimental setup. On a more philosophical note, the degree to which a paragraph or document with only 10% watermarked content even constitutes "a watermarked document" anymore is somewhat context dependent, and from a practical standpoint it is not clear a deployed detector should always return "AI" or "watermarked" in that scenario. However, we aimed to explore the limits of detection techniques in our study.
>
>
> > How does the performance of watermarking compare to other detection methods in detecting machine-generated text within shorter sequences or fragments?
>
> In Section 4.4 we conduct a comparative study of detection algorithms and find that watermarking and the other detection algorithms offer varying degrees of reliability in both the machine paraphrase and the copy-paste settings, but that in extreme scenarios like CP-1-10%, the WinMax approach is the best of the group.
>
> In our intial experiments we utilized DetectGPT as a representitive of the stongest post-hoc detector available at the time, but throughout the rebuttal period we have run an additional experiment to incorporate a newer detector called RADAR. This model is an adversarially trained classifier designed to be a reliable detector of model generated content even after paraphrasing and we find it to be quite competitive with the Retrieval approach. However, as demonstrated clearly in the updated ROC chart analysis, both Retrieval and RADAR do not achieve the same levels of sensitivity in the low FPR range of 0.001 to 0.1 that are achievable with watermarking.
>
> > Could watermarking be used in conjunction with other detection methods to enhance overall detection reliability, particularly in complex and adversarial settings?
>
> Of course! We believe that hybrid approaches to AI generated content detection that simultaneously leverage techniques like watermarking, loss based methods like DetectGPT, trained ML classifiers like RADAR, and database approaches like the Retrieval method, are likely to offer system performance levels that are unachievable with each approach alone, especially across varied domains and use cases. We leave this very interesting and important avenue of research to future work.

---

> ### Author Response · Authors · 2023-11-17
> **Continued discussion and list of updates to manuscript**
>
> > I would like to express my appreciation for the efforts made by the authors in addressing the comments and incorporating improvements into the manuscript. It is admirable to witness the dedication to enhancing the paper despite the initial concerns.
>
> Thank you! We appreciate the continued feedback and feel that the new experiments and changes made to the manuscript so far have increased the quality and completeness of the work.
>
> > However, some of my concerns still persist to some extent. Firstly, I would appreciate it if the authors could delve deeper into providing theoretical insights. Specifically, it would be beneficial for the readers to understand the key takeaways and insights that they can apply to their adjacent work. Clarification on what sets this work apart and how the proposed approach stands out would be valuable. The current response appears somewhat high-level, and additional detail in this regard would be highly beneficial.
>
> Concerning how our research relates to other theoretical work in the literature, we would like to direct the reviewer to Appendix Section A.2 which discusses the relationship between our reliability results and other theoretical work on the "impossibility of detection". A particular point we draw attention to through our study is the claim that a watermark can simply be removed by a paraphraser that samples from the set of text with the same content. Such theoretically optimal paraphrasers have, so far, not been demonstrated to exist. Our experiments using _real_ paraphrasing models such as a prompted ChatGPT instance or a purpose tuned paraphrasing model, suggest that strong watermarks are reasonably robust to these types of transformations despite existing analytic results on the asymptotic difficulty of detection.
>
> Further, theory established in Kirchenbauer 2023, and consolidated in the opening of Section 4, predicts that the type of watermark we study should exhibit favorable scaling behavior in detection performance as a function of tokens of evidence. We find that these watermarks do exhibit a predictable growth in detection power as a function of text length, despite strong attacks, but that not all of the post-hoc detection baselines we considered enjoy this property. We establish this "detectability @ T" analysis as an important evaluation methodology with which to conduct future work on watermarking and detection.
>
> To tie all of this together, we would like to succinctly describe the contributions that we make in this work:
> - We optimize both the watermarking process and the detection algorithm to be more robust to realistic text corruptions.
> - We show that watermarks retain useful levels of detectability despite realistic paraphrasing attacks like passing through a strong language model like ChatGPT.
> - We develop a simultaneously controlled but realistic type of text mixing attack to explore what impact "copy and pasting" fragments of AI generated and human written content together has on watermark strength.
> - We conduct a human study that shows that not even expert paraphrasing is sufficient to totally scrub a watermark from a piece of text.
> - Finally, by comparing the reliabilty of advanced watermarks to alternate detection methods like loss-based and retrieval approaches, we find that watermarks do in fact provide a level of reliabilty that other methods do not, as predicted by prior theory.
>
> > Secondly, I am eager to read the revised version of the paper. Could you kindly confirm whether the new version has been uploaded? I am particularly interested in reviewing the highlighted changes to gain a clearer understanding of the modifications made.
>
> Since the initial reviews were released, we have updated the submission PDF (currently available at the link above) in three main ways:
> 1. In response to a request by reviewer o62T, we have incorporated experiments on a third post-hoc detector, RADAR. These new results are incorporated into Figures 5 and 6 along with the prose in section 4.4 "A Comparative Study of Detection Algorithms". Some further details on the new detector model have been added to Appendix Section A.11.3 "Baseline Method Details and Hyperparameters".
> 2. In response to another request from reviewer o62T concerning the impact of the more advanced SelfHash watermark we study on the utility of a model, we have added a new set of experiments measuring TriviaQA performance under watermarking. Theses are tabulated in Appendix Section A.6.
> 3. In response to your request for a more extensive discussion on wider implications of watermarking and detection technology, we have greatly extended our ethics statement to include this new material, which can be found in Section 7.

---

> > ### Comment · Reviewer_jTXD · 2023-11-20
> > **Final comment**
> >
> > I express gratitude to the authors for facilitating my comprehension of the work. Following a thorough examination of the rebuttal message, I have revised my review from a negative to a positive stance. It is justifiable to acknowledge the authors with commendation for their diligent efforts in producing a robust work and providing thoughtful responses. However, it is pertinent to note that I lack expertise in this domain, thus I leave the discussion open to other reviewers and ACs. I posit that research within the domain of LLMs-enabled applications is a worthwhile pursuit, as exemplified by the authors' endeavors in this regard. Good luck and happy holiday!

---

### Official Review · Reviewer_o62T · 2023-10-27

**Soundness:** 3 good
**Presentation:** 2 fair
**Contribution:** 3 good
**Rating:** 6
**Confidence:** 4

**Summary:**

This paper studies an important and timely topic on the robustness of watermarked AI-generated text detection, in the scenarios of language model paraphrasing, human paraphrasing, and partial editing (e.g., copy and paste).

**Strengths:**

1. The research scope (whether an AI-general text embedded with watermarks can be easily removed or not) is an important and timely topic.
2. Empirical results are abundant and show the promise of the reliability of the evaluated watermark methods

**Weaknesses:**

1. Probably due to page limits, most spaces are used for presenting numerical results. The methodology section, including a new watermark method (e.g. SelfHash) and a new detection method (WinMax) in Sec. 3 is relatively short (roughly one page), and much important information is deferred to the Appendix.
2. The analysis will be more complete if it includes more recent and advanced post-hoc detection methods (such as RADAR https://arxiv.org/abs/2307.03838), because DetectGPT is known to be non-robust to paraphrasing. I would like to see the result of RADAR in ROC analysis and with varying token lengths.

**Questions:**

1. For the analysis of post-hoc detection, given that DetectGPT is shown to be fragile against paraphrasing (Sadasivan 2023), can the authors add a new analysis with a more robust post-hoc AI-text detector, like RADAR? I would like to see the result of RADAR in ROC analysis and with varying token lengths, as in Fig. 5 and Fig. 6.

2. I don't see much discussion on the utility of the studied watermark methods (especially the new one, SelfHash). if a watermark is robust but lacks usefulness, it is impractical. Can the authors report the usefulness of the new watermark methods that have been not studied in (Kirchenbauer 2023)?

---

> ### Author Response · Authors · 2023-11-17
> **Response to Reviewer o62T**
>
> We appreciate the reviewer's acknowledgement that page limits forced us to make some difficult decisions about what to showcase in the main body of the work. We tried to make the pointers to more details where required as easy to follow as possible for the reader.
>
> In direct response the reviewer's request, we have performed an experiment with the publicly available RADAR model, and have incorporated the results into Figure 5 and Figure 6. We find that RADAR is significantly more reliable than DetectGPT in our evaluation setup and agree with the reviewer that this improves the quality of the comparitive study between detection methods substantially.
>
> The edits necessary to incorporate the RADAR technique are mostly confined to main body section 4.4 "A Comparative Study of Detection Algorithms" and appendix section A.11.3 "Baseline Method Details and Hyperparameters".
>
> We would like to be transparent about a few caveats. First, running the RADAR model on Llama generations to keep parity with the rest of the results constitutes a slight domain shift since we are using RADAR-Vicuna (only checkpoint available), but this transfer pair was quite favorable according to Figure 3 in their paper so we believe the setup is still fair. Second, the token limit of the model (512) means that for the copy-paste experiments, the model didn't receive all 1000 available tokens like the other detector approaches, though this has less of an impact on the GPT and Dipper based paraphrase experiments because those sequences average only 300 and 800 tokens after paraphrasing, respectively. For RADAR, the curves stop at 512 for this reason.
>
> Next, concerning the utility of SelfHash, we did not forsee major changes in utility, compared to the hashing scheme employed in Kirchenbauer 2023 (only changes in reliability), but your review raises the valid point that we did not explicitly verify this. To address this concern, we have now added a utility study, comparing the more advanced SelfHash scheme evaluated in this work to the original LeftHash scheme, measuring their impact on a model's ability to generate free-form answers to TriviaQA questions. We use slightly more modern chat models and generate watermarked responses for the first 1000 TriviaQA questions marking an answer as correct if the string of the ground truth answer, or one of its aliases, is included in the model's response. The models we evaluate are LLaMA-2-chat-7b and Zephyr-7b-beta.
>
> In the updated draft, we tabulate these results in Appendix Section A.6. Overall, we find that the utility of the model is only minimally impacted by the SelfHash watermark scheme, with the original LeftHash watermark yielding similar results. This is in line with our expectations that the two schemes should behave similarly under this analysis. We thank the reviewer for the suggestion and feel that this improves the completeness of our evaluation.

---

> > ### Author Response · Authors · 2023-11-21
> > **Any questions regarding our additional experiments and manuscript update?**
> >
> > We just wanted to reach out and see whether you had any remaining questions regarding the additional experiments run in response to your review requests? The updates to the manuscript are described in our response above and the PDF viewable at the submission link is current with respect to these additions.
> >
> > Again, we appreciate your constructive suggestions and do believe the work has been strengthened in the process, thanks!

---

> > > ### Comment · Reviewer_o62T · 2023-11-21
> > >
> > > I thank the authors for their efforts in addressing my concerns and updating the manuscript. I've increased my score accordingly.

---

### Official Review · Reviewer_o8L2 · 2023-10-31

**Soundness:** 3 good
**Presentation:** 4 excellent
**Contribution:** 3 good
**Rating:** 6
**Confidence:** 5

**Summary:**

The paper studies the robustness of language watermarks against paraphrasing attacks. These attacks involve both paraphrasing models and tests against human rephrasing. The surveyed watermark is shown to be robust against all attacks. It is also more effective than retrieval-based attacks in cases where watermarked text is inserted into non-watermarked text. Comprehensive experimental details are provided in an appendix by the authors.

**Strengths:**

* Overall, I quite liked the paper and think that it addresses an interesting problem.

* The appendix is extremely detailed and offers a lot of valuable information on the reproducibility of their study.

* The paper shows that the studied text watermark is robust against many attacks, including human paraphrasing, which is somewhat surprising.

* The paper outlines many useful parameters and graphs for evaluating the robustness of watermarking.

**Weaknesses:**

The main issue I have with the paper is an unclear threat model: What is an attacker allowed to do to paraphrase sequence correctly? When is a paraphrased text too dissimilar from the watermarked text? The paper does not answer these fundamental questions, but follow-up papers must rely on these answers to propose improved attacks.

Consider the following example: A human and a paraphraser want to preserve the "meaning" of the watermarked text. The watermark hides with high probability in high-entropy token sequences, such as names, locations, and numbers. Is a paraphraser allowed to replace these with random other names and locations, or would that be considered "unsuccessful" paraphrasing? I would love to see that discussed in the paper so that future papers can meaningfully improve on the presented attacks.

**Questions:**

* What is a successful paraphrasing attack?

* What does an attacker need to achieve to undermine a watermark's robustness?

---

> ### Author Response · Authors · 2023-11-17
> **Response to Reviewer o8L2**
>
> We'd like to address the reviewer's concerns by first clarifying that our intention with this research was to conduct an empirical study into the _reliability_ of watermarking techniques in realistic scenarios rather than robustness to a targeted attack. The formal definition of a threat model necessitates an attacker and defender relationship that is stronger than the setup we studied and further, tends to require a well defined notion of the attacker's "budget" which we believe is difficult to define in this case.
>
> Nevertheless, we did our best quantify whether participants produced valid paraphrases during our human study by utilizing an established metric from the literature on paraphrase semantic similarity called Paragram-SP (Weiting, 2022). Other work has has shown that the average similarity values of human written paraphrases are around 0.76 under this metric (Krishna, 2023), and thus we considered anything above a 0.7 to be a valid paraphrase. All our annotators achieved at least this average similarity to the starting text, despite significant reductions in watermark strength in their paraphrases, and we spot checked the responses to confirm no obvious circumventions of the paraphrase task guidelines. We provide further details about the study setup and instructions in Appendix A.13. One could build a bounded threat model around such a similarity measure, however, since it is a model-based score, the generality of such a bound or budget might be limited.
>
> Finally, as a remark on entropy, one can draw a connection between the amount of available entropy given a prompt and whether many paraphrases of a correct answer or response exist. If a valid answer to a query necessitates the use of specific entities likes names and locations, then 1) a soft watermark is unlikely to change whether or not those entities are generated, and 2) a paraphraser is unlikely to be able to come up with a valid paraphrase that doesn't include said entities. Low entropy token spans are not regions where the watermark will be densely embedded, and thus whether or not a paraphraser changes them is likely not a determining factor in resultant watermark stength after the paraphrase. We think that the reviewer might find the new results in Appendix Section A.6 concerning TriviaQA (prepared in response to a separate review question) quite illustrative of the correspondence between entities and entropy. This experiment supports the hypothesis that the watermark is relatively "inactive" when the model is generating short, entity-centric responses to factual questions.

---

> > ### Comment · Reviewer_o8L2 · 2023-11-18
> > **Thank you for your clarifying remarks.**
> >
> > Thank you for your responses!
> >
> > I am still wondering about the longevity of these research results. As I understand it, one reason to study robustness in the adversarial setting is that its security properties extend to the non-adversarial setting that you study. While it is true that _current_ generation of machine paraphrasing systems are not able to paraphrase and remove the watermark, future paraphrasers might be able to remove the watermark (even without trying), invalidating your results. In other words, while some insights from the paper are surprising and meaningful in the long term, such as the inability of human experts to remove the watermark, why is it meaningful to study _weak_ removal attacks to claim reliability of a watermark?
> >
> > I agree that there is no "perfect" score that captures similarity between text for any context, and I appreciate that the authors evaluate other datasets in A.6. However, there exist no "perfect" score in any ML field, but we agreed on some scores that show meaningful improvement (e.g., CLIPScore or the Fréchet Inception Distance in the image domains). Since your paper is fully dedicated to benchmarking one watermarking method _in depth_, I was hoping that you aim to set standards for future work in this field by proposing a threat model.
> >
> > Thank you for the link to reference A.6. This illustrates the point made above in that you lack a definition of "correct" paraphrase. For example, in Table 4 of your Appendix, you have the following prompts (Prompt 4):
> >
> > > As an expert copy-editor, please rewrite the following text in your own voice while ensuring
> > that the final output contains the same information as the original text and has roughly the
> > same length. Please paraphrase all sentences and do not omit any crucial details. Additionally,
> > please take care to provide any relevant information about public figures, organizations, or
> > other entities mentioned in the text to avoid any potential misunderstandings or biases.
> >
> > Any paraphrase or human who follows the task will retain the watermark if the text contains a few high-entropy token sequences (e.g., names or addresses). These may not be "factual", i.e., the model may have made them up, which is why they have a high entropy. Any paraphrased text that mentions a combination of many high-entropy token sequences will be detectable with your watermark and it is impossible to remove it, because the paraphrasing instructions are flawed.
> >
> > A better instruction would have been to add something like "Please find new mentions for all non-public figures, organizations, or
> > other entities not essential to the text." in the case where you randomly sample from a model.

---

> > > ### Author Response · Authors · 2023-11-20
> > > **Response regarding the motivations and goals of our study**
> > >
> > > We appreciate the comments from the reviewer and agree that there is room for work that establishes threat models and family of metrics in the style of L_p bounds and FID scores, which are a ubiquitous combination in the study of adversarial attacks and defenses in CV. In fact, there are certain other works that have begun some of this work though they do not specifically concern watermarking. Kumar 2023 describes a threat model and certified defense against a family of prompt perturbation attacks on LLMs, and Kandpal 2023 studies a threat model for backdoor attacks in the modern era of generative language models capable of in-context learning.
> > >
> > > However, this was not our goal with this research. For context, when we first designed our study we were responding to an initial outpouring of interest and and discussion in the community regarding the practical usefulness of watermarks like the one that Kirchenbauer 2023 proposed. We observed that many people already had strong intuitions about the problem. Some were convinced that such a watermark was trivially brittle, and would be naturally removed from generated text, even if the text was used in practical scenarios including the following:
> > > 1. Paraphrasing by a commercially available large language model like ChatGPT. Many people use these models daily, and it is natural step to use these models to refine, rewrite and paraphrase existing generated text.
> > > 2. Manual rewriting of the text. Anyone with reasonable writing proficiency will usually take the output from a generative model and modify its outputs to include their writing style, interests and knowledge.
> > > 3. Mixing generated text into human-written text, or text from any other source. This is also a very natural procedure one could employ when utilizing text generated by a language model. The text may be inserted into other existing text, or used to replace only a few key sentences in a larger document.
> > >
> > > The primary goal of our research was to take these three hypotheses, formalize them into controlled experiments, and run realistic tests with tools as representative as possible of the real world scenarios these hypotheses described. In this way, our research is a "usability" study or analysis of watermark behavior "in-the-wild"/"in-deployment", leaving strictly adversarial investigations to other work.
> > >
> > > We see this experimental design as valuable for future work. New watermarks, or other tools to detect machine-generated text, can be tested against these case studies to investigate their reliability in practical scenarios that involve the use of machine-generated text that might be embued with a watermark.
> > >
> > > Further, a few clarifications on the other comments in your response:
> > > - Regarding weakness of removal attacks, we used a strong language model, ChatGPT and a purpose trained paraphrasing model Dipper, which is a T5 variant specifically developed in another study on paraphrasing as an attack on watermarking.
> > > - Prompt 4, used throughout the work, was the best performing prompt with respect to watermark removal for the ChatGPT model we utilized. However it was not part of the instructions to human paraphrasers during the human study. The instructions to annotators were different (A.13.5), and we remark that they gave the annotators information concerning presence of a watermark in the text.
> > > - On your suggestion for a better paraphrase instruction, we note that starting watermarked passages in the human study were responses to ELI5 questions, not generic webtext completions, so the space of possible valid responses and entities, and their valid paraphrases was relatively bounded. That said, technically, we knew our annotators could have trivially replaced all entities and factual information with other tokens, or even written completely unrelated paragraphs in their place. They would have achieved a higher watermark degradation than they did using this technique. However, this would have negatively impacted the semantic similarity to the original text (as captured by P-SP), and we informed them that this would be checked for their paraphrases.

---

> > > > ### Comment · Reviewer_o8L2 · 2023-11-22
> > > >
> > > > Thank you for your reply. I will keep my current positive score.

---

### Author Response · Authors · 2023-11-17
**To all reviewers**

We would like to thank the reviewers for all the time and effort they have put into the reviewing process. We are pleased to see that the thoroughness of our empirical evaluation was recognized by all, and we have done our best to run additional experiments requested and refine parts of the manuscript to better align with reviewer suggestions. We respond individually to the concerns and questions of each review in detail below.

---

### Author Response · Authors · 2023-11-22
**Thanks again**

We would like to say a final thank you to the three reviewers for participating in an engaging and fruitful discussion of our work over the course of the last week. Thank you in advance for your continued effort in the rest of the review process, and for those who are celebrating it, have a wonderful Thanksgiving!

---

### Meta-Review · Area_Chair_DPct · 2023-12-08

**Metareview:**

The paper studies the reliability of the LLM watermarking scheme of Kirchenbauer et al. to paraphrasing.
The paper introduces a new "selfhash" method, and then evaluates it against human and LLM-based paraphrasing.
The results indicate that current watermarks are fairly reliable against simple forms of paraphrasing.
As noted by some reviewers, it is not quite clear what these results mean in the long-run as LLM paraphrasers get better, or against attackers who might employ more involved attack strategies.
But the experiments presented are solid and should provide a good baseline for future work.

**Justification For Why Not Higher Score:**

The paper has some solid experimental evaluation, but it is unclear how generalizable these findings are.

**Justification For Why Not Lower Score:**

It's a decent paper studying a relatively narrow question, but it executes this well.

---

### Decision · Program_Chairs · 2024-01-16

Accept (poster)